

# Oceanographic dataset collected during the 2021 scientific expedition of the Canadian Coast Guard Ship *Amundsen*

Tahiana Ratsimbazafy[1,3], Thibaud Dezutter[1,3], Amélie Desmarais[1,3], Daniel Amirault[1,3], Pascal Guillot[1,2,3], and Simon Morisset[1,3]

[1]Amundsen Science: 1045 Avenue de la Médecine, Pavillon Alexandre-Vachon, Local 3432, Université Laval, Québec, QC, G1V 0A6, Canada
[2]Québec-Océan: 1045 Avenue de la Médecine, Pavillon Alexandre-Vachon, Local 2078, Université Laval, Québec, QC, G1V 0A6, Canada
[3]These authors contributed equally to this work.

**Correspondence:** Tahiana Ratsimbazafy (tahiana.ratsimbazafy@as.ulaval.ca; amundsen.data@as.ulaval.ca)

**Abstract.**

Since 2003, the state-of-the-art Canadian Coast Guard Ship (CCGS) research icebreaker *Amundsen* furrows the Canadian Arctic waters to support novel research endeavors and collect oceanographic data. This paper presents the data acquisition, the processing methods and an overview of the data collected during the 2021 expedition as the ship traveled over 30 000 km

during 122 days across the Canadian Arctic Ocean, collecting sea surface, atmospheric and seabed underway measurements. A total of 266 casts of a conductivity, temperature and depth profiler mounted on a rosette (CTD-Rosette) were also conducted to monitor the main physical, chemical and biological parameters of the water column. More specifically, the data here presented were collected with the CTD-Rosette across historical sampling transects in Davis Strait, the North Water Polynya (NOW), and Cape Bathurst. A 182 km dedicated survey of Moving Vessel Profiler (MVP), equipped with CTD, transmissometer, dissolved

oxygen, fluorescence, sound velocity sensors, was conducted across Hudson Strait. We also present an overview of the data collected by the underway systems (seabed, thermosalinograph and atmospheric). Such data are essential in understanding the impacts of climate warming on the unique environments of the Canadian Arctic Ocean. Amundsen Science supports and promotes easy access and sharing of such valuable data to the scientific community.

## 1   Introduction

The Canadian Arctic Ocean covers 4 million km$^2$, and over 70% of Canada's coastline is within the Arctic region (Niemi et al., 2020). Such a vast area embraces numbers of unique ecosystems, on which it has been challenging to establish a global, strong, and reliable baseline to fully understand the impact of global warming. Over the years, ice camp (Untersteiner et al., 2009; Massicotte et al., 2020), research vessels (such as the CCGS *Amundsen* and the CCGS *Louis Saint-Laurent*), Inuit Knowledge (Weatherhead et al., 2010; Gearheard et al., 2010), and moorings (Armitage et al., 2020; Dezutter et al., 2021;

Nadaï et al., 2021) have all been tools used to acquire data and to fill gaps in the Canadian Arctic Ocean knowledge. Meanwhile, satellites and meteorological stations have been providing reliable and rigorous data sets of sea ice and air temperature since



the industrialization period. Previous work showed that the Arctic is warming twice as fast as the rest of the world, and will continue to do so (Serreze et al., 2009; Gulev et al., 2021). In the Canadian Arctic, the summer sea ice extent has been reduced by 7.5% per decade over the 1968-2020 period (Environment and Canada, 2021a) in addition to global thinning (Lindsay and
Schweiger, 2015; Meier et al., 2014), while the air temperature has been increasing between 1.5-3.7 $^\circ C$ from the 1948-2018 period over the Arctic land (Environment and Canada, 2021b). The unique ecosystems of the Arctic Ocean, driven by extreme seasonality in light regime and sea ice cover, which dictate the energy transfer through the food web, from micro-algae to large marine mammals (Falk-Petersen et al., 2007) have been directly and indirectly impacted by the drastic loss of sea ice. Changes in timing of biological events (Niemi et al., 2020; Hauser et al., 2017), northern shift of species (Dunmall et al., 2018) (Higdon
and Ferguson, 2009) along with increase marine traffic (Halliday et al., 2022; Johnston et al., 2017; Dawson et al., 2018) and pollution (Adams et al., 2021) are all known consequences of the climate change and ice-retreat in the Canadian Arctic.

Since 2003, the CCGS icebreaker *Amundsen* and its leading-edge scientific instrumentation has been monitoring the Canadian Arctic by supporting dozens of large-scale national and international research initiatives from academia, local communities, and the public and private sectors. The CCGS *Amundsen* is the only coast guard ship seasonally dedicated to science, and
over 1400 refereed scientific publications and 400 datasets have resulted from the CCGS *Amundsen* expeditions. Hosted at Université Laval, Amundsen Science is the organization responsible for the management of the scientific mandate of the research icebreaker CCGS *Amundsen*. Specifically, Amundsen Science manages the vessel's pool of scientific equipment, coordinates the deployment of the icebreaker for science, and provides logistical, financial, and technical support to user programs. In order to promote and share Canadian Arctic oceanographic data, the objective of this paper is to present an overview of datasets col-
lected by the core instruments managed by Amundsen Science over the 2021 Amundsen scientific expedition. The expedition took place from July 4th to November 3rd and was divided into five Legs[1] taking place in the Labrador Sea, the Baffin Bay, the Canadian Arctic Archipelago, and the Beaufort Sea (Fig. 1). Eight multidisciplinary national and international research programs took part in this annual expedition (Table B1). Research programs objectives are central to the Amundsen expeditions and are supported by Amundsen Science through oceanographic data acquisitions: detailed CTD-Rosette, Moving Vessel
Profiler (MVP), underway measurement of sea surface properties with Thermosalinograph (TSG) and atmospheric data, multi-beam echo sounder, single beam echo sounder and sub-bottom profiler datasets were collected during this expedition and are presented in this paper. Information on additional samples collected by the research teams onboard during the 2021 expedition can be obtained by contacting the respective principal investigators (as detailed in Table B1) or by consulting the expedition report (Amundsen Science, 2021b).

---

[1]A Leg is a determined period during which the sampling operations of the ship are scheduled.



**Figure 1.** Map of the CCGS *Amundsen* 2021 expedition, including ship track and the locations of CTD-Rosette stations and MVP casts. (©ESRI 2023).

## 1.1 Regional settings

### 1.1.1 Labrador Sea and Hudson Strait

The Labrador Sea, located between Labrador and Greenland, receives several water currents from different part of the Arctic regions. The Baffin Current from the Baffin Bay brings cold water as well as drifting icebergs and ice islands into the Labrador Sea, acting as corridor for ice transport (Yang et al., 2016; New et al., 2021). Such moving ice blocs was reported by Marko et al. (2014) as presenting risks for activities and operations conducted offshore Newfoundland. Cold waters from Hudson's Bay Current circulates through Hudson Strait to the Labrador Sea where it joins warm sub-arctic water from the North Atlantic



Current (McGeehan and Maslowski, 2012; Yang et al., 2016) and feeds the source of water mass formation in the Labrador and Irminger Seas (Kieke and Yashayaev, 2015).

Exchanges of carbon dioxide, oxygen, and heat between the deep ocean and the atmosphere have been reported by Clarke and Coote (1988); Azetsu-Scott et al. (2003); DeGrandpre et al. (2006); Körtzinger et al. (2008); Yashayaev and Seidov (2015) to occur in Labrador Sea area. They arise from deep ocean convection process during winter when heat losses occurs and cold waters sink to greater dept, produces the Labrador Sea Water (McCartney, 1992; Kieke et al., 2009), and allows the possibility of exchange between the intermediate and deep waters with the surface water and the atmosphere (Kieke et al., 2009).

Such complexity of system processes occurring in the Labrador Sea demonstrate how important is the gathering of scientific knowledge as a guidance for different decision makers to monitor risks associated to different activities such as exploration and exploitation of oil and gas in order to keep the safety of marine ecosystem in the region.

### 1.1.2 Baffin Bay

Located between the Baffin Island and Greenland, Baffin Bay is connected to the Lincoln Sea through Nare Strait one of the pathways of sea ice transport from the Arctic Ocean Sea into Baffin Bay (Kwok, 2005). Lancaster and Jones Sound are considered as corridor as well, connecting Baffin Bay to the Arctic Ocean allowing ices and fresher waters being transported downstream to Davis Strait and the Labrador Sea (Tang et al., 2004).Despite it's location in high latitude, Baffin Bay is under the influance of the northward West Greenland Current (WGC), bringing warm and salty waters and exchanges heat, salt, and ice with the southward Baffin Island Current in an anti-clockwise circulation (Arctic Monitoring and Assessment Programme (AMAP), 2018).

In Nares Strait, between Ellesmere Island and Greenland, ice arches typically form each winter at both its northern and southern end (Moore et al., 2021). The formation of an ice arch in Nares Strait and the resulting cessation of ice transport, the input of warm and salty Atlantic water from the West Greenland current, and upwelling of warmer waters, all contribute to the formation of the North Water (NOW) Polynya, a year-round expanse of open waters in Smith Sound and northern Baffin Bay which is the largest and most productive of its kind (Melling et al., 2001; Moore et al., 2019). The thinning of the Arctic sea-ice is negatively affecting the stability of these ice arches, which results in an acceleration in the loss of multi-year ice from the Arctic and could impact the NOW polynya ecosystem (Moore et al., 2021; Kwok et al., 2010; Moore et al., 2019).

Glaciers, icecaps, and icefields of the Canadian Arctic and Greenland also contribute to the freshwater input in Baffin Bay via their runoff through fjords, rivers, and calving (Bamber et al., 2018). Some icefields, such as the Manson icefield (southeast Ellesmere Island, Canada) directly connect with Northern Baffin Bay. In the Canadian Arctic, the Manson icefield has the highest concentration of surging glaciers, characterized by surging periods and variable glacier and freshwater flow (Copland et al., 2003).

During the 2021 Arctic expedition, the CCGS *Amundsen* visited Baffin Bay and Northern Baffin Bay for scientific sampling activities to monitor seawater physics, chemistry nutrients, contaminants, and the biodiversity present along precise historical transects, glacier fronts and Baffin Island fjords.





### 1.1.3 Canadian Arctic Archipelago (CAA)


The CAA covers several islands and channels between Banks Island in the west and Baffin and Ellesmere Islands in the east. This area is characterized by the mixing of Pacific, Atlantic and Arctic-originated waters, a strong year-to-year variability in oceanographic and biological processes and changing sea ice conditions (Michel et al., 2006; Pizzolato et al., 2014). Over the last decades, marine transport in the region has increased due to the easing in shipping navigability driven by reductions in

the extent, thickness and age of sea ice (Kwok et al., 2009). As the CAA could become the larger outlet for Arctic Ocean ice area loss (Howell and Brady, 2019), risk of collision resulting in potential Arctic oil spills are high (Helle et al., 2020). Arctic Council's Working Group on the Protection of the Arctic Marine Environment (PAME) (2020) indicated the importance of collecting bathimetric and sub-bottom data for seafloor mapping, identifying potential geohazards and obstacle to a safe navigation in the newly open Arctic.

Collecting bathymetric data for seafloor mapping is a challenging task for a large area with limited access such as the Arctic. An earlier version of digital bathymetric data, International Bathymetric Chart of the Arctic Ocean (IBCAO) was first introduced by Jakobsson et al. (2000) at AGU conference on 1999. The data was part of the declassified historic sounding collections from the United State and the British submarines between 1957 to 1988. Several updates has been made since using models and new data from different sources Jakobsson et al. (2008, 2012).

During the expedition year 2021, the CCGS *Amundsen* collected echo-sounding data at Smith Bay, Ellsmere Island. Descriptions of instruments for seabed data acquisition and samples from the seabed mapping are presented in Sect. 2.3.3 and Sect. 3.5 respectively.

### 1.1.4 Beaufort Sea

Recently, important changes on sea ice has been reported in both Beaufort Sea and the Mackenzie Shelf region of the Arctic
Ocean (Comiso, 2002; Galley et al., 2016; Mudryk et al., 2018). The first year ice is becoming more common in the area due to increased transport of old ice away from the regional peripheral Arctic seas (Nghiem et al., 2007; Ogi et al., 2008; Hutchings and Rigor, 2012). The Beaufort Sea shelves are mostly influenced by input sediment-rich from the Mackenzie River

The Beaufort Sea is characterized by a broad shelf onto which the Mackenzie River indicated in (Carson et al., 1998) as the most sediment-rich river in the Arctic by then with $130 \times 106$ tonnes/yr, carries a large and highly seasonally variable amounts
of freshwater (Carmack and Macdonald, 2002).

Along the Mackenzie Shelf stretches the highly productive Cape Bathurst polynya, the third biggest expanse of open water that has existed year-round (Arrigo and van Dijken, 2004). This ecosystem is also exceptional since it provides habitat for some of the highest densities of birds and marine mammals in the Arctic (Harwood and Stirling, 1992; Dickson and Gilchrist, 2002), although climate change-induced stress and anthropogenic activities are putting and increasing pressure on the ecosystem
(Hoover et al., 2021).

Since 2002, extensive multidisciplinary research programs have been conducted in the Beaufort Sea area from the CCGS *Amundsen* through international overwintering research programs including Canadian Arctic Shelf Exchange Study (CASES)



in 2003-2004 and Circumpolar Flaw Lead (CFL) Study in 2007-2008. Environmental and oceanographic research activities were also conducted in the offshore region of the Mackenzie Shelf, shelf slope and Beaufort Sea within the framework of the

Beaufort Regional Environmental Assessment program (BREA).

## 2 Data acquisition, processing and quality control

Data collected and the subsequent methods applied to ensure quality assurance and quality control (QAQC) are presented in the following section. Unless specified otherwise, all processing scripts are publicly available at Guillot et al. (2022).

### 2.1 CTD-Rosette profiler

The CCGS *Amundsen* is equipped with a Sea-Bird Scientific© SBE 911plus CTD (SBE 9plus CTD unit combined with SBE 11plus V2 deck unit) mounted on a SBE 32 carousel water sampler with twenty-four 12L Niskin bottles, model 110BES by Ocean Test Equipment (Appendix C2). Along with the basic SBE CTD sensors (temperature, conductivity, pressure, dissolved oxygen), the SBE 9plus supports several auxiliary sensors (nitrate, transmissometer, colored dissolved organic matter, fluorescence, photosynthetically active radiation (PAR)). Conductivity, temperature and fluorescence sensors are doubled to ensure

data redundancy and reliability. A surface PAR (SPAR) sensor is also installed at the top of the bridge between 10-15 m from the sea surface and provides reference values. The main sensors are factory calibrated every year before the cruise period and are post-calibrated after the cruises that allows to detect any problems such as drift. An independent LADCP (lowered acoustic doppler current profiler) is also mounted to the frame in order to measure the horizontal current velocities throughout the water column. The LADCP is a Teledyne 300kHz instrument and is looking downward. It is set up to record data with 8 m bin size,

one ping per second and zero blanking. Technical specifications related to each sensor mounted on the rosette are summarized in Table 1.

The water samples collected with the CTD-Rosette are used by multiple scientific programs (Table B1 in Appendix) and by Amundsen Science for data validation. The conductivity sensors are compared with salinity water samples analyzed with a 8400B Guildline salinometer. The dissolved oxygen sensor is validated with water samples analyzed via the Winkler method.

Other sensors such as the Suna nitrate sensor or the Seapoint fluorescence sensors are validated with water samples if they are analyzed and available from other scientific teams.

The Fig. 2 depicts the temperature and salinity anomalies recorded by the dual sensor system. Good agreement is obvious between the two temperature sensors. However, the salinity sensor presents a slight and continuous drift by the end of Leg 5. The comparison with the water samples allowed to determine the faulty senor.

The CTD data are processed using the Seabird data processing software and using the recommended processing module sequences. The QAQC procedure is mainly based on the Global Temperature and Salinity Profile Program (GTSPP) Commission et al. (2010) quality control tests. Additional tests including pump status checking or downcast versus upcast comparisons or doubled sensors comparisons (temperature and salinity) are applied as well. Moreover, the main sensors such as temperature, conductivity and dissolved oxygen are factory post-calibrated to detect potential issues.



**Table 1.** Instruments and specifications of the CTD-Rosette deployed during the 2021 expedition of the CCGS *Amundsen*.

| Instrument | Company | Variables | Specifications |
|---|---|---|---|
| SBE 3plus | Sea-Bird Scientific | Temperature | Resolution at 24 Hz: 0.0003°C |
| | | | Initial accuracy: $\pm$ 0.001°C |
| SBE 4 | Sea-Bird Scientific | Conductivity | Resolution at 24 Hz: 0.00004 Sm$^{-1}$ |
| | | | Initial accuracy: 0.0003 Sm$^{-1}$ |
| Deep Suna | Sea-Bird Scientific | Nitrate | Range: 3000 $\mu$M |
| | | | Resolution: $\pm$0.5 $\mu$M |
| | | | Accuracy: $\pm$2 $\mu$M |
| SBE 43 | Sea-Bird Scientific | Dissolved Oxygen | Range: 120 % of surface saturation |
| | | | Accuracy: 2 % of saturation |
| FLCDRTD | Sea-Bird Scientific (Wetlabs) | CDOM | Range: 0 to 500 ppb |
| | | | Sensitivity: 0.09 ppb |
| C-Star | Sea-Bird Scientific (WetLabs) | Beam Transmittance | Optical pathlength: 25 cm |
| | | | Wavelength: 25 657 nm |
| | | | Sensitivity: 1.25 mV |
| | | | Response time: 0.167 s |
| Digiquartz® pressure | Paroscientific, Inc. | Pressure | Range: 0-6800 m |
| | | | Resolution at 24 Hz: 0.001 % |
| | | | Initial accuracy: 0.015 % |
| SCF | SeaPoint Sensors, Inc. | Fluorescence | Range: 0-15 $\mu$gL$^{-1}$ * |
| | | | Sensitivity: 0.33 V$\mu$g$^{-1}$L |
| | | | Minimum Detectable Level: 0.02 $\mu$gL$^{-1}$ |
| QCP-2350 | Biospherical Instrument Inc. | PAR (Irradiance) | Spectrum: 400-700nm |
| | | | PAR range: 0-5000 $\mu$Einsteins m$^{-2}$ s$^{-1}$ |
| QCR-2200 | Biospherical Instrument Inc. | SPAR (Surface Irradiance) | Spectrum: 400-700nm |
| | | | PAR range: 0-5000 $\mu$Einsteins m$^{-2}$ s$^{-1}$ |
| LADCP | Teledyne RDI Workhorse | Current velocities | Frequency at 300 kHz |

\* The SCF can be used in four sensibility and range configurations. The 15 $\mu$gL$^{-1}$ range configuration was used for all casts of the 2021 expedition. For QCP-2350 specifications, see Biospherical Instruments Inc. (2014). Most QCR specifications are known to be identical with those of QCP sensors.

The conservative temperature and absolute salinity presented in this paper are calculated using the Gibbs SeaWater (GSW) Oceanographic Toolbox (McDougall and Barker, 2011), based on the thermodynamic equation of seawater 2010 (TEOS-10) (IOC et al., 2010).

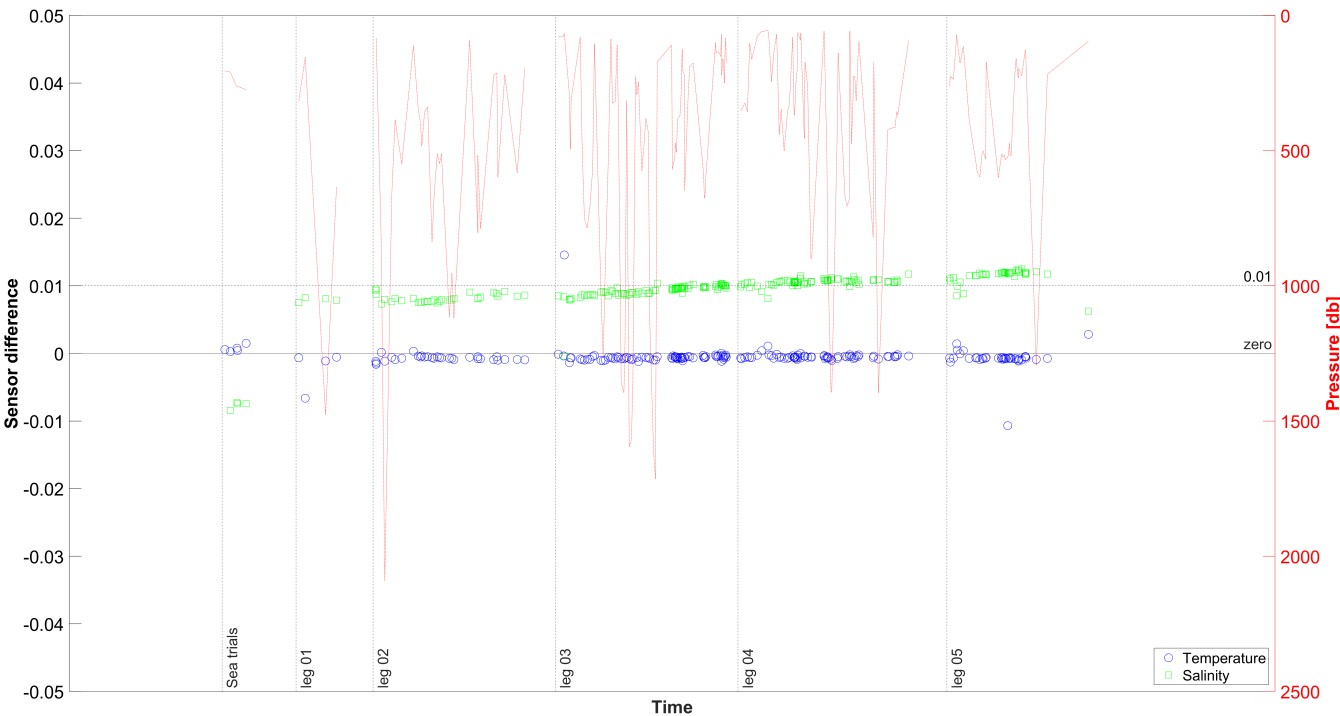

**Figure 2.** CTD anomaly evolution between the dual Sea-Bird Conductivity (green) and Temperature (blue) sensors along the sea-trial and the 5 Legs. Both Salinity and Temperature were used to calculate the anomaly. They are extracted from the highest pressure values (red) from each profiles (last quarter of the pressure range).

## 2.2 Moving Vessel Profiler ®

The AML Oceanographic Moving Vessel Profiler® (MVP300) is part of the central pool of instruments operated by Amundsen
Science on the CCGS *Amundsen*. With its capability of supporting both shallow and deep-water data sets collection, the MVP's primary function is to allow accurate data collection of all the water column without the need to stop the vessel. The system includes a computer-controlled smart winch and deployment system that allows continuous up and down casts of the free fall fish. The fish is equipped with seven sensors to record oceanographic data along a transect of vertical dives of the fish. The installed probes include temperature, conductivity, pressure, sound velocity, dissolved oxygen, fluorescence, and transmittance
sensors. Technical specifications related to the sensors mounted on the MVP are presented in the Table 2.

MVP data processing is conducted using a MATLAB® in-house processing script developed by Amundsen Science. Processing steps are sequentially applied on each cast of a given MVP transect and include the conversion of analog inputs into digital inputs, the flagging of out of range and spiking values and the average over 1m depth.

Derived parameters are also calculated (e.g.: salinity, sound velocity, etc.) and manual data check is conducted to ensure the
quality of the data. The final processed data are saved in text-based format. More details are available in the MVP processing report (Amundsen Science (2021a)).





**Table 2.** Instrumentation and specifications of the MVP during the 2021 expedition

| Instrument | Company | Variables | Specifications |
|---|---|---|---|
| Micro CTD | AML | Temperature | Range: -2-32 °C |
| | | | Initial Accuracy: 0.005 °C |
| | | | Resolution: 0.001 °C |
| | | Conductivity | Range: 2-70 mScm$^{-1}$ |
| | | | Initial Accuracy: 0.01 mScm$^{-1}$ |
| | | | Resolution: 0.0015 mScm$^{-1}$ |
| | | Pressure | Range: 0-600 bar |
| | | | Initial Accuracy: 0.05 %FS |
| | | | Resolution: 0.1 dbar |
| Micro SV | AML | Sound velocity | Range: 1375-1600 ms$^{-1}$ |
| | | | Initial Accuracy: 0.05 ms$^{-1}$ |
| | | | Resolution: 0.01 ms$^{-1}$ |
| | | Pressure | Range: 0-600 bar |
| | | | Initial Accuracy : 0.05 %FS |
| | | | Resolution: 0.1 dbar |
| Rinko III | JFE Alec | Dissolved Oxygen | Range: 0-100 % |
| | | | Response time: 0.9 s (90%) |
| | | | Drift: 5 % per month |
| ECOFLO | Sea-bird (WetLabs) | Fluorescence | Range: 0-125 $\mu$gL$^{-1}$ |
| | | | Sensitivity: 0.062 $\mu$gL$^{-1}$ |
| | | | Wavelength: 470 and 695 nm |
| C-Star Transmissometer | Sea-Bird (WetLabs) | Beam Transmittance | Optical pathlength: 25 cm |
| | | | Wavelength: 657 nm |
| | | | Sensitivity: 1.25 mV |
| | | | Response time: 0.167 s |



## 2.3 Underway systems

### 2.3.1 Thermosalinograph (TSG)

The CCGS *Amundsen* is equipped with a continuous underway seawater sampling system including a ThermoSalinoGraph
(TSG) for temperature and conductivity measurements coupled with a fluorometer, a sound velocity sensor and an additional
temperature sensor. A 20 mesh size strainer is placed at the beginning of the line to filter the surface seawater, which circulates
under the ship to reach the TSG installation under the vessel's deck before entering the water line. Technical specifications
related to the sensors included on the TSG are presented in Table 3. Geolocations (latitude, longitude), dates and times of the
recording are presented alongside the Practical Salinity, and Sea Water Temperature. Other variables related to the ship (vessel
speed [kn], water flow rate [L/min]) are presented in the dataset as well.

The configuration of the TSG is shown in appendix C. Data processing is conducted using the MATLAB® in house pro-
cessing script (available at Sect. 6), and requires the use of navigation data, atmospheric data, seawater surface salinity samples
from the TSG (analyzed with Guildline salinometer) and CTD-Rosette data for inter-comparison and correction. Processing
steps include the flagging of out of range values, the averaging over 1 minutes and intercomparison and correction with collo-
cated data from the CTD-Rosette and in-situ samples is made. The final processed data are saved in text-based format. More
details are available in the TSG processing report published with the dataset (Science (2021c)).

**Table 3.** Instrumentation and specifications of the Thermosalinograph (TSG) during the 2021 expedition (Morisset, 2021)

| Instrument | Company | Variables | Specifications |
|---|---|---|---|
| SBE 45 | Sea Bird | Temperature | Range: -5-35 °C |
| | | | Initial Accuracy: 0.002 °C |
| | | | Resolution: 0.0001 °C |
| | | Conductivity | Range: 0-7 Sm$^{-1}$ |
| | | | Initial Accuracy: 0.0003 Sm$^{-1}$ |
| | | | Resolution: 0.00001 Sm$^{-1}$ |
| | | Salinity (derived value) | Initial Accuracy: 0.005 psu |
| | | | Resolution: 0.0002 psu |
| SBE 38 | Sea Bird | Temperature | Range: -5-35 °C |
| | | | Initial Accuracy: 0.001 °C |
| | | | Resolution: 0. 00025 °C |
| WETStar | Wetlabs | Fluorescence | Range: 0.03-75 $\mu$gL$^{-1}$ |
| | | | Initial Accuracy: 0.03 $\mu$gL$^{-1}$ |
| Micro X P1S0 | AML | Sound velocity | Range: 1375-1625 ms$^{-1}$ |
| | | | Initial Accuracy: 0.025 ms$^{-1}$ |
| | | | Resolution: 0.001 ms$^{-1}$ |

Data quality assessment for the practical salinity is shown on Figure 3. Uncertainty value is roughly equal to 0.01 PSU while the bias stays constant when compared to the result from the CTD Rosette. To be consistant of the data sets presented in the present paper, the Absolute Salinity is presented instead of the Practical Salinity. The Absolute Salinity was calculated using the new thermodynamique equation of seawater (TEOS-10) (McDougall et al., 2009) within the Gibsbs-SeaWater (GSW) toolbox by McDougall and Barker (2011).

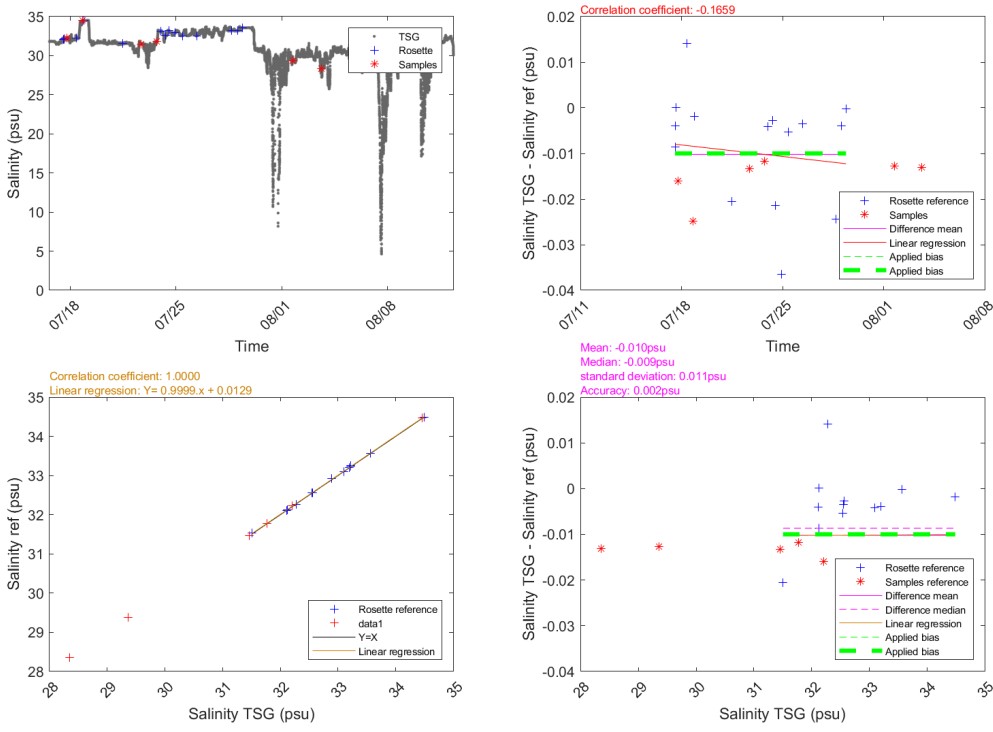

**Figure 3.** Practical Salinity quality assessment. (Morisset, 2021)

### 2.3.2 Atmospheric data

The CCGS *Amundsen* is part of the worldwide Voluntary Observing Ship (VOS) scheme, led by the World Meteorological Organization (WMO). As part of this program, an Automated Voluntary Observing Ship system (AVOS) is installed by Environment and Climate Change Canada (ECCC) on the ship to record continuous data on atmospheric pressure, wind speed, wind direction, air temperature and humidity. Amundsen Science retrieves and processes the available AVOS data and add them to the available pool of data. Data processing is conducted using a MATLAB® in house processing script (available at Sect. 6) and includes flagging out of range values, environmental corrections (e.g.: wind speed, vessel pitch and roll, shadow, etc), averaging data over 2 minutes and manual checks.





Processed data are saved in text-based format (one file for each leg). More details are provided in the AVOS processing report (Science (2021a)).

**Table 4.** Instruments and specifications of the AVOS underway system

| Instrument | Company | Variable | Specifications |
|---|---|---|---|
| Digital Barometer -PTB-210 | Vaisala | Atmospheric pressure | Range: 50-1100 hPa |
| | | | Accuracy: 0.35 hPa |
| | | | Resolution: 0.1 hPa |
| Anemometer - 05103 | Young R.M. | Wind speed | Range: 0-100 m s$^{-1}$ |
| | | | Initial accuracy: 0.3 ms$^{-1}$ |
| | | Wind direction | Range: 0-360° |
| | | | Initial accuracy: 3° |
| MP101 | Rotronic Meteorological | Air temperature | Range: -40-60°C |
| | | | Initial accuracy: 0.2°C |
| | | Humidity | Range: 0-100% RH |
| | | | Initial accuracy: 1% RH |
| SPAR - QCR2200 | Biospherical Instruments | Photosynthetically active radiation | Range: 0-5000 $\mu$Em$^{-2}$s$^{-1}$ |
| | | | Initial accuracy: 1.7 $\mu$Em$^{-2}$s$^{-1}$ |

### 2.3.3    Seabed Data Acquisition Instruments

During the 2021 mission, the CCGS *Amundsen* was equipped with a Kongsberg® multibeam echo sounder (EM302), and a Knudsen 3260 sub-bottom profiler to acquire dedicated and underway seabed data. The seabed data acquisition systems

opportunistically operate throughout the entirety of each scientific expedition. Seabed mapping relies on analyzing acoustic pulses propagated and received by echo sounders to generate georeferenced models of the seabed. The bathymetry consists of three-dimensional topographic point clouds of the seafloor, whereas sub-bottom consists of vertical profiles of strata layers below the seafloor. Arctic seabed mapping supports a wide range of research (habitat mapping, geology, paleoclimatology, glacial history, archaeology, geohazards, geopolitics, and more) and provides ships with needed information to navigate safely.

The CCGS *Amundsen*'s EM302, upgraded to the Kongsberg® EM304 MKI in summer 2022, is a 30kHz system capable of acquiring data in 10-8000 m water depth. The multibeam echo sounder integrates Global Navigation Satellite System and inertial measurement unit data from the Applanix POSMV V4 to georeference generated soundings. The Kongsberg Seafloor Information System (SIS)] acquisition software retrieves acquired EM302 data from the system's processing unit and optimizes data quality using various parameters and tools applied by onboard technicians. The following procedures are used to process

raw multibeam data and convert it into deliverable products:





- Sound velocity profiles are derived from each CTD-Rosette station and applied to the multibeam during acquisition. Onboard operators integrate real-time sound velocity measurements at the transducer face, and conduct quality assurance of each cast. If CTD-Rosette casts are sparse and require long transits between deployments, synthetic sound velocity profiles are applied using the 2009 and 2013 World Ocean Atlas Models (WOA09 and WOA13).

- Raw EM302 data are imported to CARIS HIPS and SIPS®, where data is georeferenced to absolute positions, depths are converted to reference the Mean Sea Level (MSL) Datum using the WebTide Tidal Solution Model®, and total propagated uncertainty is calculated.

- Operators revise data in the HIPS and SIPS® 2D and 3D Subeditor tool and manually reject erroneous data points from the data set.

- Cleaned bathymetry data is exported to raster surfaces, catalogued, and distributed.

Processed data are saved in raster format and publicly available through a multitude of sources indicated on Amundsen Science's website Science (2023).

## 3  Data overview

### 3.1  CTD-Rosette

During the 2021 expedition, Amundsen Science deployed the CTD-Rosette on 266 occasions, producing respectively 4, 35, 83, 103 and 41 casts during legs 1 to 5 spanning across the Canadian and Greenlandic Arctic, from the coast of Labrador to Beaufort Sea (Figure 1). Historical CTD transects were visited during the 2021 expedition and three were selected to present an overview of the CTD data collected during this expedition in Davis Strait, northern Baffin Bay (NOW), Cape Bathurst. These transects are called *historical*, as they were visited multiple times since 2003 by the CCGS *Amundsen*. Note that the

Davis Strait transect has only been visited since 2018, but is intended to be visited yearly from now on. For each transect, interpolated values of Conservative temperature (CT), the dissolved oxygen (DO) and absolute salinity (AS) are presented. The seafloor data represented in the following figures are retrieved from echo sounder depth data along the ship track. TS diagrams compiling all the profile from the transect are also presented for each transect in order to describe water mass of the area.

### 3.1.1  Davis Strait

Nine stations were sampled along the Davis Strait between August 17th-20th 2021 (Table A1). A clear distinction between water properties of the eastern and western part of Davis Strait was observed (Fig.4). The Davis Strait water masses have been extensively described in the past and the four main water masses of the region were observed (Fig. 5) during the sampling from the CCGS *Amundsen* (Tang et al., 2004; Curry et al., 2011; Lehmann et al., 2019; Punshon et al., 2014). The West

Greenland Slope Current (WGSC) water mass is characterized by discontinued layers of highest salinity (up to 34.5 g/kg), warm



temperature (2.4°C) and relatively poor DO2 concentration (250-300 $\mu M$). This water mass is present along the Greenland slope between 100-700 dbar. The water mass present over the Greenland Shelf, is characterized by the highest temperatures recorded along the transect (up to 6.5 °C), relatively rich in oxygen (300-340 $\mu M$) and fresher waters (up to 34.1 g/kg). Over the western side of the Strait, Baffin Island Current (BIC) water mass, can be observed in the top 200 dbar. This water mass

is characterized by the lowest salinity recorded in this transect (between 30-33.7 g/kg) and highest DO2 concentration (420 $\mu M$) in the surface layers, which extends over the Greenland slope. Coldest temperature are also observed in the BIC (-1.5°C), between 50-200 dbar. Finally, the Atlantic derived waters of the western part of Davis Strait are characterized by relatively warm ($\leq$ 2°C), oxygen depleted and relatively salty (> 33.7 g/kg), (< 280 $\mu M$) waters below 300 dbar (Fig. 9b, Fig. 4c, and Fig.4d). Water masses distinctions in Fig. 5 were derived from Curry et al. (2011); Tang et al. (2004).

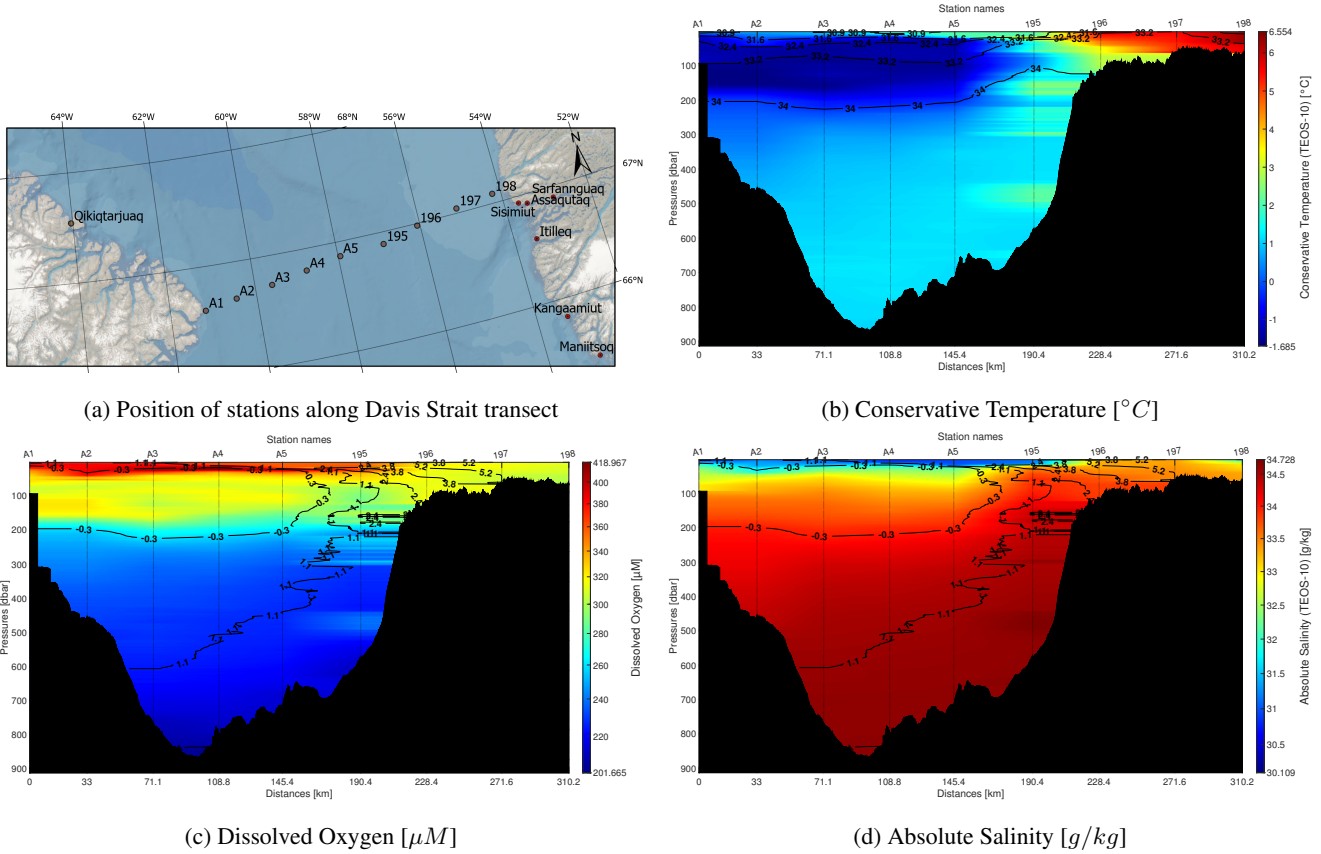

(a) Position of stations along Davis Strait transect

(b) Conservative Temperature [$^\circ C$]

(c) Dissolved Oxygen [$\mu M$]

(d) Absolute Salinity [$g/kg$]

**Figure 4.** Davis Strait transect profiles during leg 3 of 2021 expedition. Isolines represents conservative temperature (isotherms, in °C).





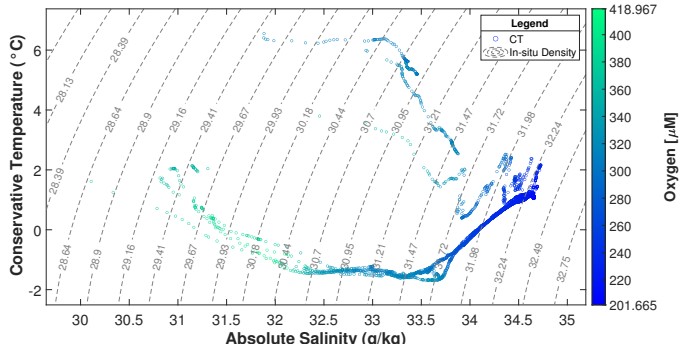

**Figure 5.** TS Diagram represented by Absolute Salinity (AS),and Conservative Temperature (CT). The isolines are the In-Situ Density calculated with AS and CT.

### 3.1.2 North Water Polynia (NOW)

Seventeen stations were sampled along the NOW Transect between August 31st and September 3rd (Table A2). Figure 6a, 6b, 6c, 6d show position of the different CTD-Rosette casts, interpolated values of conservative temperature (CT), dissolved oxygen (DO) and absolute salinity (AS) respectively through the water column at each stations along the NOW transect, . As stated by (Bâcle et al., 2002; Lobb et al., 2003), the West Greenland Current (WGS) brings the warm and salty Atlantic waters in the region between 200 and 400 dbar. This water mass can be observed in Fig. 6 by two distinctive cells between stations 107 and 111 and between station 111 and 116, where the maximum value of AS (34.5 g/kg) and lowest value of DO concentration (< 250 $\mu M$) of the transect are located. Relatively warm temperature of 1°C also define the water mass. This water mass is trapped under a cold halocline observed between 50-150 dbar Bâcle et al. (2002); Lobb et al. (2003). Based on Figure 6d, this layer is characterized by absolute salinity values ranging from 33-33.8 g/kg, temperature colder than 0.7°C and waters relatively rich in DO,ranging from 295-340$\mu M$. A warm, fresh and DO rich (max 400 $\mu M$) polar mixed layer (PML) is observed from surface at roughly 75 dbar along the transect. Between stations 101 and 107 a colder and fresher surface layer is observed over the ridge (Fig. 6d). A thin rich DO layer can be observed within the PML around 30 dbar and is extended up to 100 dbar over the ridge (stations 101 to 105). This water is mostly coming for the north by Smith Sound and from the West by Jones Sound and has been warmed up during summer (Bâcle et al., 2002; Lobb et al., 2003).

All CTD casts of NOW transect are plotted in TS-diagram (Figure 7).Different water masses signatures showcased through the diagram are derived from (Bâcle et al., 2002). Along the transect, two water masses defined by two sets of conservative temperature, on the left side of the figure. They correspond to density values less than 29.47 kg·m$^{-}$3. Mixture of waters are observable in the layer with density values between 29.47 kg·m$^{-}$3 and 31.4 kg·m$^{-}$3.

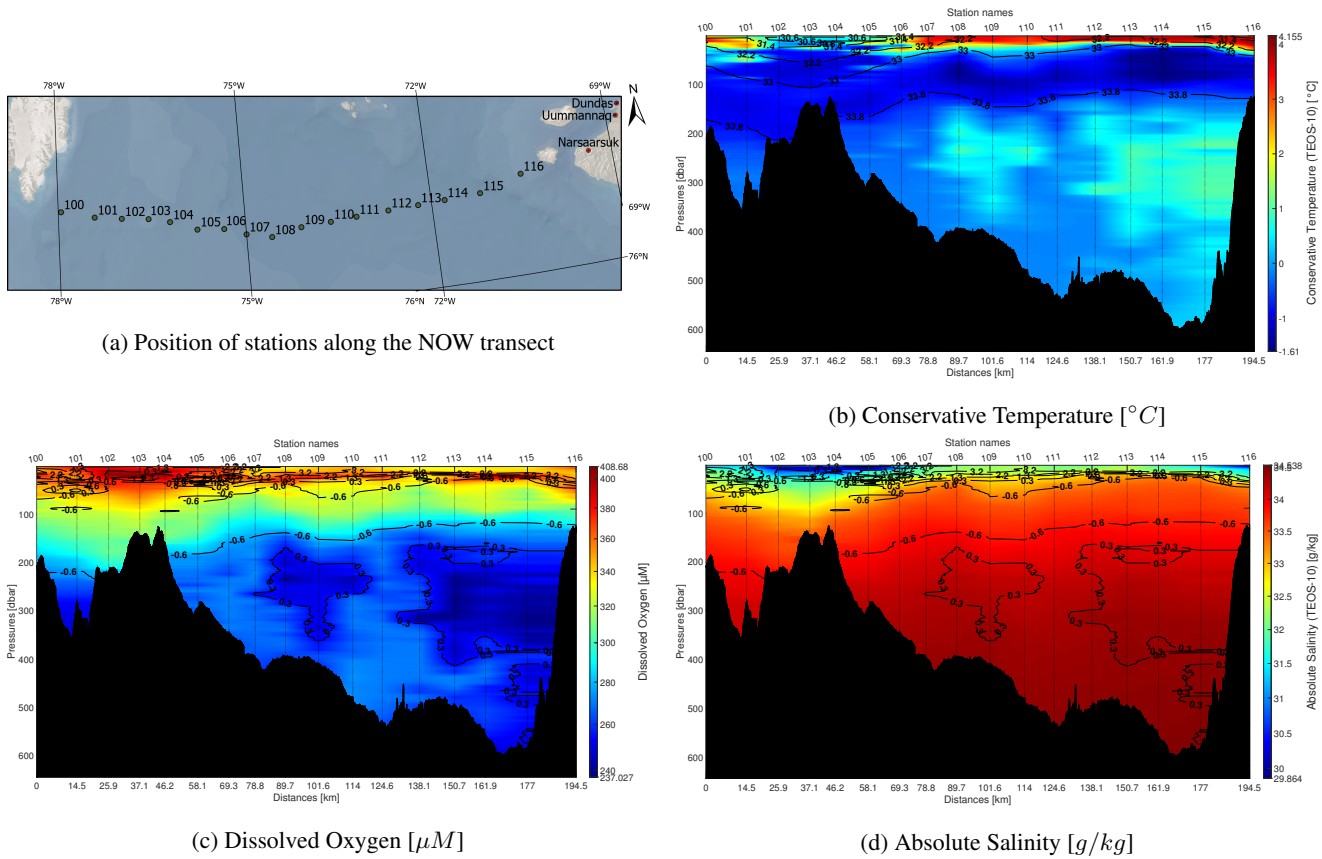

(a) Position of stations along the NOW transect

(b) Conservative Temperature [$^\circ C$]

(c) Dissolved Oxygen [$\mu M$]

(d) Absolute Salinity [$g/kg$]

**Figure 6.** NOW transect profiles during leg 3 of 2021 expedition. Isolines represents conservative temperature (isotherms, in $^\circ$C).

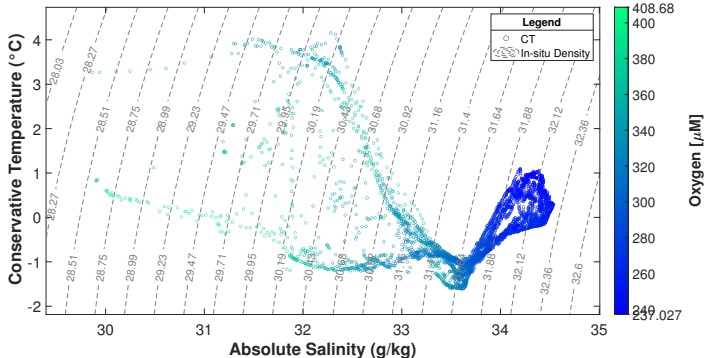

**Figure 7.** NOW TS Diagram. The isolines are the in-situ density calculated with absolute salinity and conservative temperature.





### 3.1.3 Cape Bathurst

Twelve stations were sampled from Sachs Harbour towards Cape Bathurst on September 18th- 19th (Table A3). The transect crosses over a major gateway of Arctic water mass exchange. TS diagram regrouping all casts along the Cape Bathurst transect displays similar profiles than previously discussed by Lansard et al. (2012); Simpson et al. (2008) in the southeastern Beaufort Sea region 8. In September 2021, 3 major water masses can be observed. The polar mixed layer (PML) is observed from surface to 25-50 dbar with a stable low salinity (27-30 g/kg), while temperature is much more contrasted (ranging from -1-3.5

°C). The surface layer become thinner as we go toward Cap Bathurst, from station 409 to station 420. Under the PML lays the halocline waters between 50-200 dbar, originated from the Pacific. This water mass shows increasing salinity from 31 to 34 g/kg toward the base of the layer. The coldest temperatures are recorded in the halocline layer (-1.5 °C) and DO of 270 $\mu M$. Finally, the Atlantic Water lays below 250 dbar with a salinity around 34 g/kg, temperature of 0.5 °C and a minimum DO concentration of 225$\mu M$. A possible upwelling can be observed at station 411 where there is intrusion of deep high salinity

water in the halocline layer. Upwelling were already observed in the region by Williams and Carmack (2008). Conservative temperature shows a pattern of inversion, as values are higher at the surface at station 409 and at the Station 415 to 420, and at pressures deeper than 150 dbar (Fig.9). The Amundsen has been sampling in the Cape Bathurst area 12 times since 2003, which produce an extensive dataset for interannual comparison studies. Recently, Massicotte et al. (2021) have compile and standardize the collected datasets from the 2009 MALINA expedition in the Beaufort Sea, in order facilitate their reusise. Such

dataset can be compare to the ones collected by the Amundsen during the 2021 expedition.

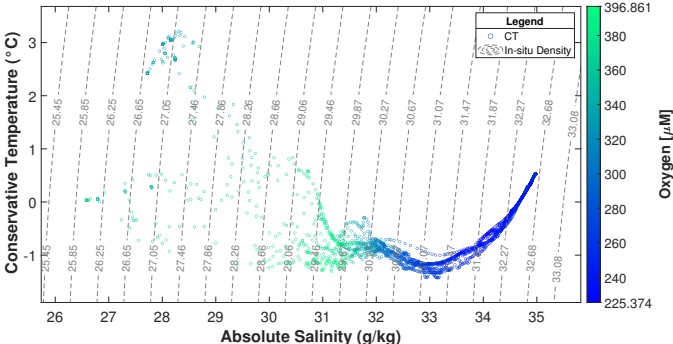

**Figure 8.** Cape Bathurst transect TS diagram. The isolines are the in situ density values calculated with AS and CT.

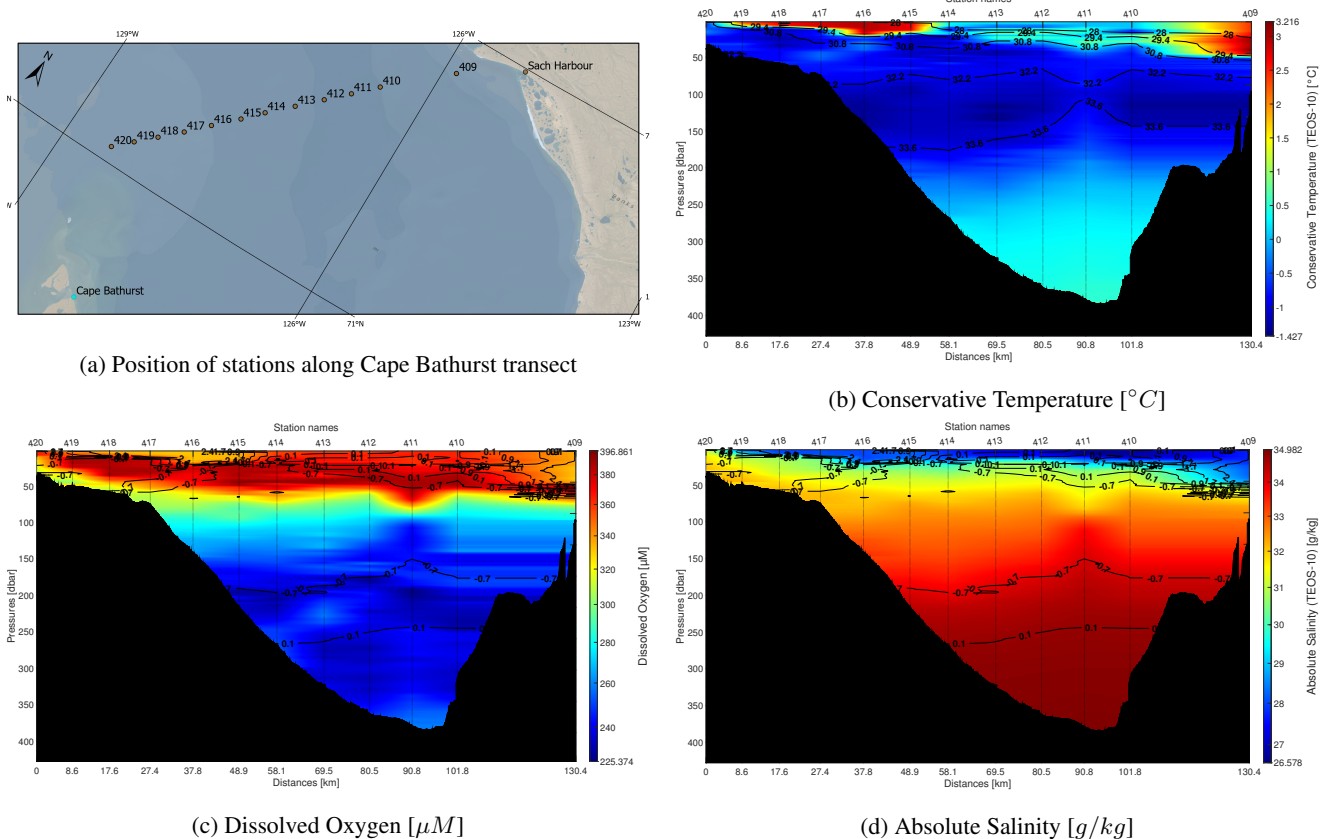

(a) Position of stations along Cape Bathurst transect

(b) Conservative Temperature [$^\circ C$]

(c) Dissolved Oxygen [$\mu M$]

(d) Absolute Salinity [$g/kg$]

**Figure 9.** Cape Bathurst transect profiles during leg 4 of 2021 expedition. Isolines represents conservative temperature (isotherms, in $^\circ$C). The north-eastern side of the transect is located on the left.

## 3.2 MVP

One MVP transect was conducted in Hudson Strait during leg 3 on August 14, 2021. A total of 197 casts (vertical dives) were carried out. The transects shown in Fig.10b and Fig.10c represent 148 casts (cast-1 to cast-148) arranged linearly from Baffin Island to the coast of Nunavik (left to right). The remaining set of casts (cast-149 to cast-197) were conducted across Diana

Bay to reach Quaqtaq. Only the first 148 casts are presented in this paper. Quality Control assessment is applied at the final step of the data processing. Uncertainties of 0.01 PSU are observed when there is relatively low vertical variability, but high vertical gradient can increase these uncertainties. For the dissolved oxygen sensor on the MVP, the uncertainty is in the order of 1%. However, it may exceed the value of 1% when high vertical temperature and salinity gradient occur.

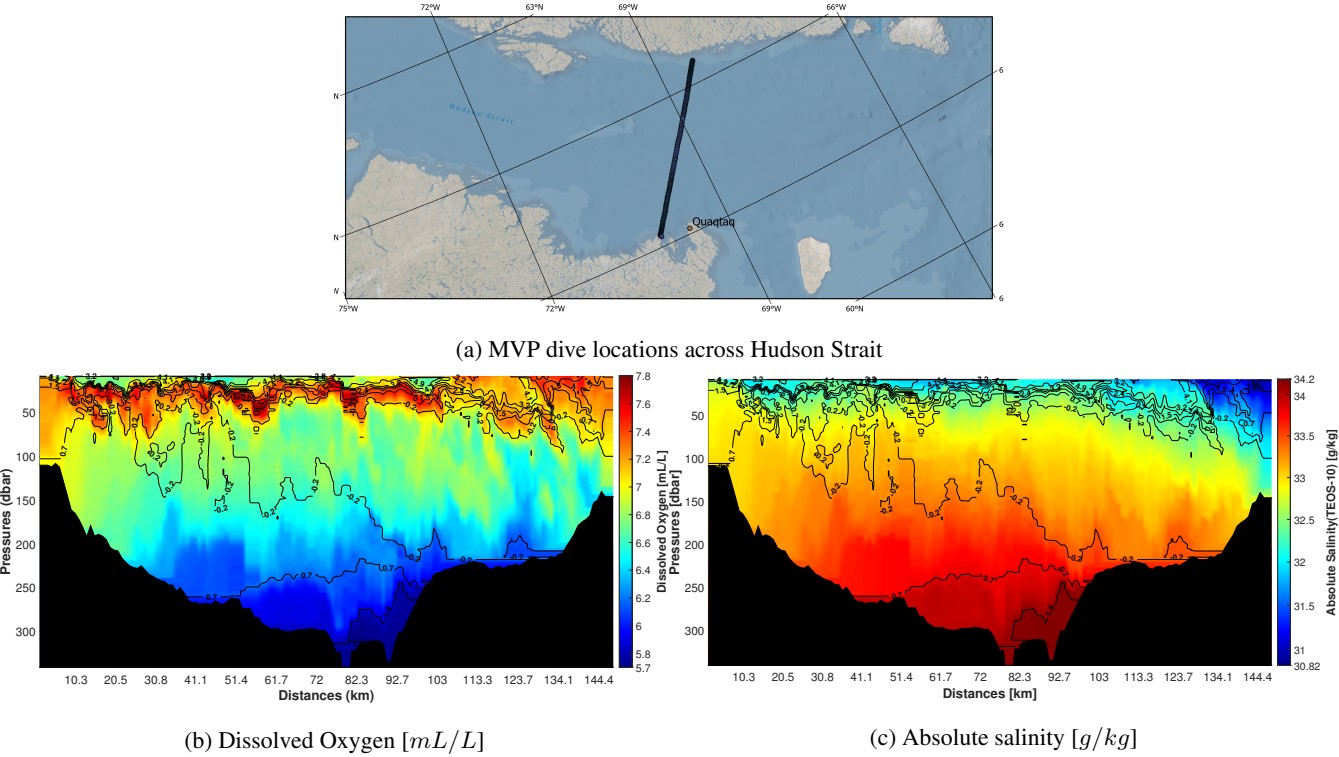

(a) MVP dive locations across Hudson Strait

(b) Dissolved Oxygen [$mL/L$]

(c) Absolute salinity [$g/kg$]

**Figure 10.** MVP section profile across Hudson Strait from leg 3 of the 2021 expedition.

Over the transect in Hudson Strait, the MVP dived approximately at each km, allowing high precision data recording. Mixing and turbulence are visible in Fig.10b, where the layer of maximum dissolved oxygen is overall well defined, but varies in depth over nearly 50 dbar. Lowest dissolved oxygen values are located in the center of the transect, below 200 dbar. Lower salinity values (below 32 g/kg) are located close to the coast of Nunavik, on the right-hand side of Fig.10c. This relatively fresh water mass is characteristic of the Hudson Bay outflow, the principal current linking the Hudson Bay System to the Labrador Sea, as described in previous studies (Ridenour et al., 2021; Straneo and Saucier, 2008). A layer of high salinity (> 34 g/kg), surrounded by the 1.5 ° conservative temperature isoline is observed at the bottom. Above this bottom layer lays a superposition of two layers of AS limited by CT values of 0.7 °C and -0.2 °C respectively. From the bottom to the surface, the variation of AS appears to be smoother compared to the variation of the DO. In general, the layer of high DO does not reflect the AS values anywhere near the surface (Fig.10c).

### 3.3 TSG

The underway thermosalinograph was operational during the five Legs of the CCGS *Amundsen* scientific expedition in 2021 but only sea surface absolute salinity (AS) derived from the practical salinity recorded during leg 2 from July 16 to August 12, 2021 is presented in 12. This subset was selected because of its high spacial variations. The minimum and maximum values of the absolute salinity ranges between 4.66-34.69 g/kg (average and median salinity of 31.04 g/kg and 31.59 g/kg respectively) in



accordance with previous observed values for Baffin Bay and Labrador Sea (Lavoie et al., 2013; Zweng and Münchow, 2006).

Lower values (fresher waters) are located in Scott Inlet and in the Clark and Gibbs Fjords (Nunavut), close to Qikiqtaaluk (Sillem) Island. Higher salinity is observed over the deeper regions of Labrador Sea and southern Baffin Bay. The close-up map in figure 12 shows the variation of the sea surface salinity at a location where fresh water masses meet the sea.

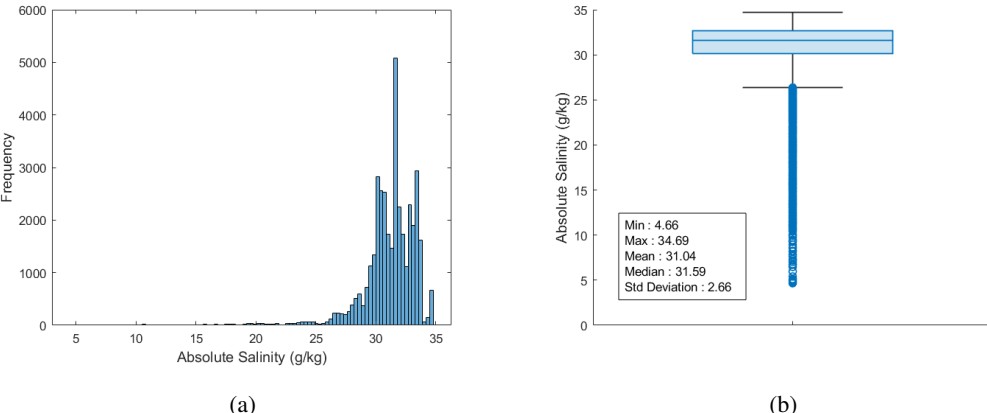

(a)                                                                    (b)

**Figure 11.** Histogram 11a and Boxplot 11b of the Absolute Salinity values from the TSG.

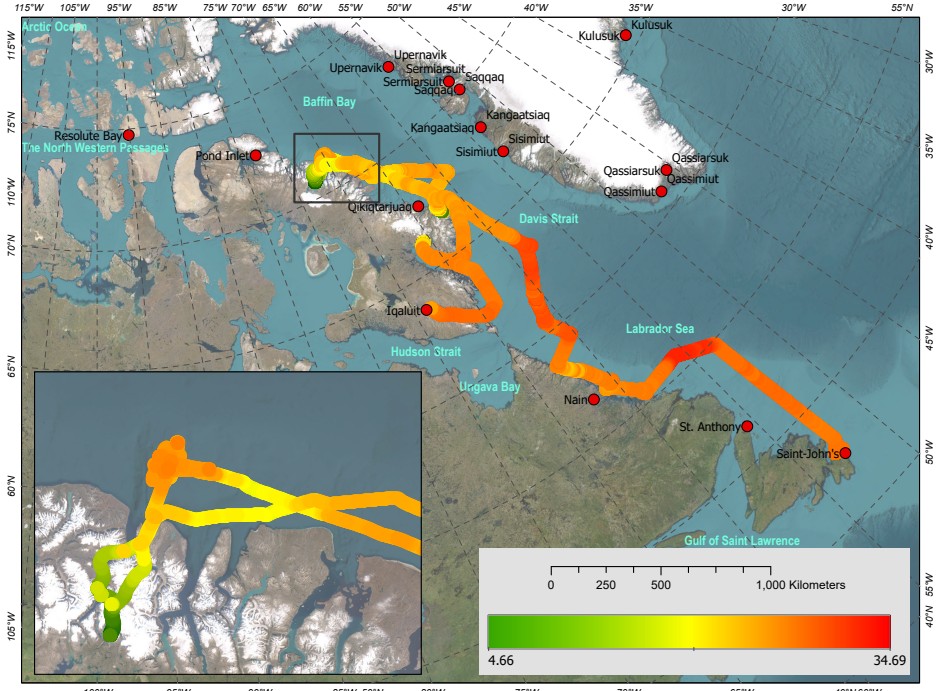

**Figure 12.** Absolute salinity (in g/kg) obtained from the underway TSG system during leg 2 of the 2021 expedition. (©ESRI 2023).




## 3.4 Atmospheric data

The Automated Voluntary Observing System (AVOS) recorded atmospheric and weather data continuously during the 2021
scientific expedition. Due to instrument malfunctions and errors identified during data processing, the percentage of data
reaching the quality standards described in Sect 2.3.2 varies between 44 % of acceptable values (relative humidity) and 97
% (atmospheric pressure), with an average of 67 % of acceptable values for all measured variables. Diurnal cycles and air
masses changes can be observed during an 8-days segment of leg 2, where the ship sailed in the Labrador Sea from offshore
Makkovik to Davis Strait (Fig. 13). A subset of four major variables (atmospheric pressure, surface photosynthetic active
radiation (SPAR), air temperature, and humidity) is presented and may be used to assess meteorological conditions during
other sampling operations and during transit. Such variables can be necessary to interpret other findings, as sea-surface physical
conditions and weather are closely related during sampling activities (Rohli and Li, 2021; Lin et al., 1996).

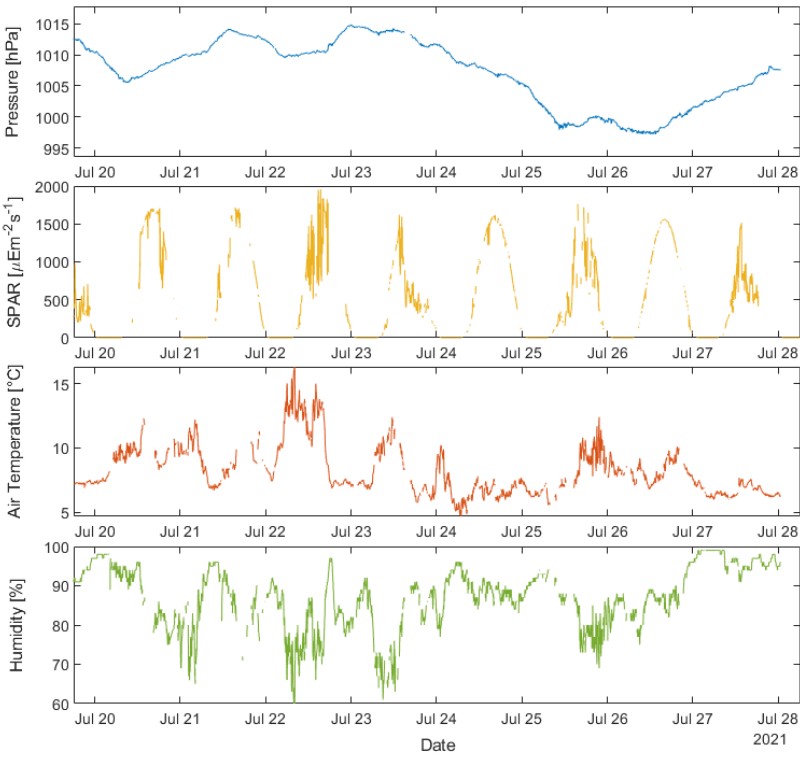

**Figure 13.** Atmospheric data from leg 2 of the 2021 expedition. Atmospheric pressure, surface
photosynthetic active radiation (SPAR), air temperature and relative humidity were measured over
20-28 July 2021.





## 3.5 Seabed Data

Over the five legs of the 2021 expedition the scientific echo sounders onboard the CCGS *Amundsen* continuously collected

data, accumulating approximately 38700 square kilometres of spatial coverage over a distance of nearly 17000 nautical miles
(Table 5).

In addition to opportunistic data collection, dedicated seabed surveys provided primary and ancillary data sets to research
projects. The following section presents two dedicated multibeam operations conducted during the 2021 Expedition. Data can
be accessed on demand at amundsen.data@as.ulaval.ca.

**Table 5.** Scientific Echosounder spatial coverage, operational distance and operational time during the 2021 expedition

| Leg | Spatial Coverage (km$^2$) | Distance (nm) | Duration (hours) |
| --- | --- | --- | --- |
| Leg 1 | 3495 | 1459 | 245 |
| Leg 2 | 9539 | 3812 | 522 |
| Leg 3 | 11135 | 3692 | 583 |
| Leg 4 | 7193 | 3894 | 470 |
| Leg 5 | 7303 | 4035 | 515 |

### 3.5.1 Makkovik

The hydrographic survey seen in Fig.14 was conducted offshore Makkovik (Nunatsiavut) during Leg 2. The survey zone is
located 12 nautical miles from the coast of Makkovik where the seafloor ranges from 100-900 m depth in transition between
coastal and deep areas, at the junction between the Aillik and Kaipokok domains of the Palaeoproterozoic Makkovik province
(Peace et al., 2018). Several overlapping geological processes take place in this province and create a complex geological

structure (Culshaw et al., 2000). The area was selected for this survey due to its topographic features and steep slopes, which
may host conditions supporting coral habitats (Edinger et al., 2011; Wareham and Edinger, 2007).

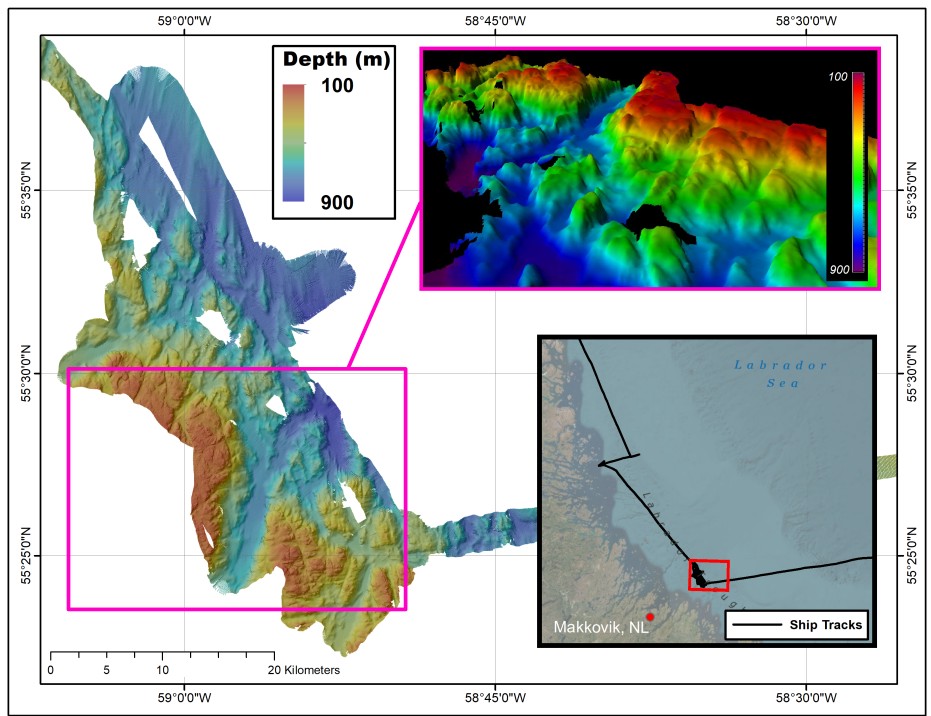

**Figure 14.** Hydrographic survey completed during leg 2 of the 2021 expedition. (©ESRI 2023)

### 3.5.2  Smith Bay

Little is known about the precise mechanisms linking the ocean-climate system and frontal positions of glaciers in the CAA (Cook et al., 2019). In order to characterise post-glacial history of Mittie Glacier and study the dynamics of glaciers in the
CAA (Amundsen Science, 2021b), bathymetric data (Fig.15) and in situ samples were collected in Smith Bay, Nunavut during Leg 3. In addition to the operation of CCGS *Amundsen* echosounders, a multibeam echosounder was installed and tested on the ship's barge along the terminus of Mittie Glacier.

Little to no seafloor data existed in Smith Bay prior to Leg 3 operations; the CCGS *Amundsen* collected the first publicly available high resolution seafloor topography data set in the area. Data depicted in Fig.15 will increase the efficiency and safety
of navigation for future operations in the area.

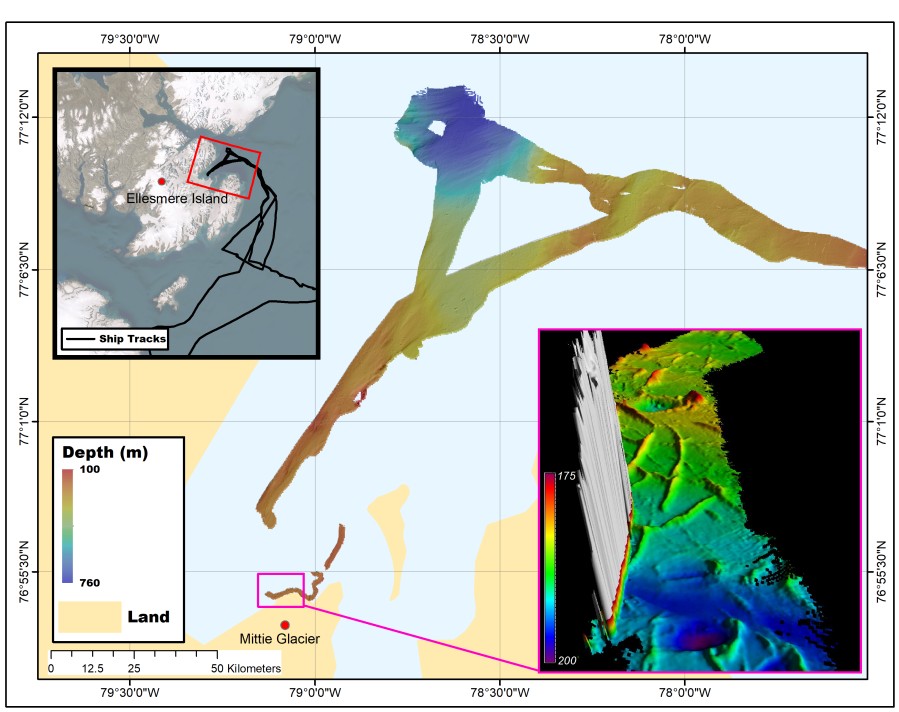

**Figure 15.** Hydrographic survey completed during Leg 3 of the 2021 expedition. (©ESRI 2023)





# 4   Conclusion

The vast amount of data collected by the central pool of equipment of the CCGS *Amundsen* during the 2021 scientific expedition in the Arctic can be used to study a vast array of research fields, from atmospheric environment to benthic ecosystems. Concurrent measurements above and under the sea surface provide invaluable tools to study the unique processes taking place in the Canadian and Greenlandic Arctic. The long-term monitoring of some of the regions of interest can allow studies of regional trends and variability. Similar research activities were also undertaken onboard other vessels in 2021 such as the USCGC Healy (McRaven, 2022) in Baffin Bay and can be used to validate or further refine findings. Only a few variable collected by the core data sets of scientific pool of equipment were presented in the paper and readers are invited to consult B1 for a comprehensive list of all data produced by the scientific community user of the CCGS *Amundsen* in 2021. Along with instruments presented in the paper, Amundsen Science is also responsible for managing a mono beam fish and zooplakton sonar (EK-80), 360 camera for ice concentration images, ROV video footage, moorings and navigation data.

Data sets collected during the scientific expedition of 2021 are archived and published in the Polar Data Catalogue for long term use. The metadata and data structures follow international standards to ensure findability and reusability of the data. Links to retrieve core data sets collected with the scientific pool of instruments onboard the CCGS *Amundsen* are available on the Amundsen Science web page, in the section Data Access at https://amundsenscience.com/data/data-access/. A complete list of instruments included onboard the CCGS *Amundsen are available at https://amundsenscience.com/canadian-research-icebreaker/scientific-equipment/.*

# 5   Data availability

*Data availability All data are available on the Polar Data Catalogue with their DOI as follow:*

- *AVOS DOI: 10.5884/12518 (Science, 2021a)*

- *MVP DOI: 10.5884/12519 (Amundsen Science, 2021a)*

- *CTD Rosette DOI: 10.5884/12713 (Science, 2021b)*

- *TSG DOI: 10.5884/12715 (Science, 2021c)*

# 6   Code availability

*Code availability All the codes used for data processing and are available on Gitub as shown below:*

- *CTD Rosette:*

- *MVP: https://git.valeria.science/amundsen/moving-vessel-profiler*

- *TSG: https://git.valeria.science/amundsen/thermosalinograph*



– *AVOS: https://git.valeria.science/amundsen/avos*



## Appendix A: Stations-ID (stations) geographical coordinates

**Table A1.** Station-ID located along Davis Strait transect.

| Station ID | Time (UTC) | Latitude | Longitude |
|---|---|---|---|
| A1 | 2021-08-17T13:25:30 | 66.6054 | -61.1941 |
| A2 | 2021-08-17T19:22:58 | 66.6692 | -60.4617 |
| A3 | 2021-08-18T04:32:52 | 66.734 | -59.6119 |
| A4 | 2021-08-18T14:53:37 | 66.8026 | -58.7708 |
| A5 | 2021-08-19T01:02:50 | 66.8697 | -57.9513 |
| 195 | 2021-08-19T10:35:01 | 66.8917 | -56.9158 |
| 196 | 2021-08-19T17:05:43 | 66.9821 | -56.0683 |
| 197 | 2021-08-19T21:53:33 | 67.0433 | -55.0851 |
| 198 | 2021-08-20T01:38:55 | 67.0842 | -54.201 |

**Table A2.** Stations-ID located along NOW transect.

| Station-ID | Time (UTC) | Latitude | Longitude |
|---|---|---|---|
| 116 | 2021-08-31T10:03:14 | 76.3802 | -70.5147 |
| 115 | 2021-08-31T15:22:38 | 76.3314 | -71.2046 |
| 114 | 2021-08-31T22:01:10 | 76.326 | -71.7856 |
| 113 | 2021-08-31T23:14:28 | 76.3205 | -72.2179 |
| 112 | 2021-09-01T01:02:36 | 76.3158 | -72.7023 |
| 111 | 2021-09-01T02:51:48 | 76.3069 | -73.2228 |
| 110 | 2021-09-01T05:35:18 | 76.2993 | -73.6364 |
| 109 | 2021-09-01T07:30:44 | 76.291 | -74.1154 |
| 108 | 2021-09-01T09:59:14 | 76.2641 | -74.598 |
| 107 | 2021-09-01T16:29:57 | 76.2835 | -75.0022 |
| 106 | 2021-09-01T19:20:46 | 76.3108 | -75.3491 |
| 105 | 2021-09-01T20:56:45 | 76.3168 | -75.7773 |
| 104 | 2021-09-02T00:09:18 | 76.3513 | -76.2076 |
| 103 | 2021-09-02T00:59:08 | 76.3686 | -76.548 |
| 102 | 2021-09-02T02:13:52 | 76.3745 | -76.9784 |
| 101 | 2021-09-02T21:38:51 | 76.3848 | -77.4143 |
| 100 | 2021-09-03T03:46:44 | 76.4105 | -77.9564 |



**Table A3.** Stations-ID located along the Cape Bathurst Transect.

| Station-ID | Time (UTC) | Latitude | Longitude |
|---|---|---|---|
| 409 | 2021-09-18T19:06:34 | 71.8689 | -125.8675 |
| 410 | 2021-09-18T23:44:42 | 71.6988 | -126.4873 |
| 411 | 2021-09-19T01:20:22 | 71.6297 | -126.7101 |
| 412 | 2021-09-19T02:30:56 | 71.5646 | -126.9189 |
| 413 | 2021-09-19T04:16:47 | 71.4954 | -127.1415 |
| 414 | 2021-09-19T06:28:18 | 71.4236 | -127.3731 |
| 415 | 2021-09-19T11:06:50 | 71.3629 | -127.5487 |
| 416 | 2021-09-19T12:15:51 | 71.2921 | -127.7697 |
| 417 | 2021-09-19T13:41:45 | 71.2248 | -127.97 |
| 418 | 2021-09-19T14:41:32 | 71.1633 | -128.1672 |
| 419 | 2021-09-19T15:58:43 | 71.1067 | -128.3449 |
| 420 | 2021-09-19T18:28:03 | 71.0517 | -128.5139 |





**Appendix B: Programs and data collected during the 2021 CCGS *Amundsen* Expedition**

Table B1: Programs and data collected during the 2021 CCGS *Amundsen* Expedition

| Programs | Leg1 | Leg2 | Leg3 | Leg4 | Leg5 | Data Collected Field of Study | Ship's Instrument | PI | Data status |
|---|---|---|---|---|---|---|---|---|---|
| Marine Spatial Planning program of Natural Resources Canada (NRCan) | X | - | - | - | - | Sediment characterization | Piston Core, boxcores, bottom camera, Multibeam, sub bottom profilers | Vladimir Kostylev (vladimir.kostylev-@canada.ca) | |
| Eastern Canada Seabirds at Sea (ECSAS) pelagic seabird surveys (EC) | X | X | X | - | - | Seabird abundance, diversity and distribution; opportunistic sightings of marine mammals and ocean pollution | | Carina Gjerdrum (carina.gjerdrum@ec.gc.ca) | Already processed (i.e., archived in database) |



*Table B1 – continued from previous page*

| Programs | Leg1 | Leg2 | Leg3 | Leg4 | Leg5 | Data Collected Field of Study | Ship's Instrument | PI | Data status |
|---|---|---|---|---|---|---|---|---|---|
| Coral seep study project (DFO) | - | X | - | - | - | (a) Benthos characterisation: Cold water corals, sponges and invertebrates abundance, distribution, and diversity. | (a) ROV video footage, gravity core, drop camera, Box Core, Baited camera | (a) David Cote (David.Cote@dfo-mpo.gc.ca) | (a) |
| | | | | | | (b) Pelagic and Sympagic POM, Nutrient and δ15N-NO3, Total inorganic carbon (TIC) and total alkalinity (TA), pCO2 and CH4. | (b) CTD-Rosette, cage for ice sampling | (b) Owen Sherwood (Owen.Sherwood@dal.ca) | (b) in progress |
| | | | | | | (c) δ13CAA and δ15N-AA signatures from sediment and zooplankton | (c) boxcore, Hydrobios | (c) Owen Sherwood (Owen.Sherwood@dal.ca) | (c) in progress |
| | | | | | | (d) eDNA - Benthic and Pelagic Community Characterization | (d) CTD-Rosette | (d) David Cote (David.Cote@dfo-mpo.gc.ca) | (d) |
| | | | | | | (e) Deepwater Faunal bioturbation and bioirrigation activity | (e) Box Core | (e) Thomas Williams (T.J.Williams-@soton.ac.uk), Guillaume Blais (guillaume.blais.8-@ulaval.ca) | (e) in progress |
| | | | | | | (f) Solenoneine analysis experiment in sediment | (f) Box Core | (f) Philippe Archambault (philippe.archambault-@bio.ulaval.ca) | (f) |
| | | | | | | (g) epifaunal colonization and settlement patterns | (g) ROV-Box Core | (g) Annie Mercier (amercier@mun.ca) | (g) in progress |
| | | | | | | (h) Sediment biogeochemistry and benthic pelagic nutrient coupling | (h) BoxCore - ROV Push Core | (h) Chris Algar (chris.algar@dal.ca) | (h) |
| | | | | | | (i) Water and sediment microbial baseline communities for potential bioremediation of an oil spill (GENICE) | (i) ROV Niskin, CTD-Rosette, box core, ROV sediment sampling (scoop + push-Core) | (i) Casey Hubert (chubert@ucalgary.ca) | (i) |
| | | | | | | (j) Mooring water and acoustic properties, organic pollutants | (j) Moorings with sediment trap, hydrophone, fish tag receiver and a semi-permeable membrane device (SPMD), current meter, CT sensor, AZFP | (j) David Cote (David.Cote@dfo-mpo.gc.ca) | (j) |





*Table B1 – continued from previous page*

| Programs | Leg1 | Leg2 | Leg3 | Leg4 | Leg5 | Data Collected Field of Study | Ship's Instrument | PI | Data status |
|---|---|---|---|---|---|---|---|---|---|
| Atmospheric methane monitoring | - | X | - | - | - | Atmospheric and dissolved methane concentration | Met Tower- CTD-Rosette | Owen Sherwood (Owen.Sherwood@dal.ca) | In Progress - Judith Vogt PhD thesis (MUN), - defending summer 2022 |
| ArcticNet-ArcticFish | - | X | X | X | - | distribution and ecology of key pelagic species in Arctic marine food webs- Fish and zooplankton | TuckerNet-Monsternet-Hydrobios-IKMT-BeamTrawl-Continuous plankton recorder (PCR)-Baited remote underwater video (BRUV) camera - EK80 echosounder-WBAT | Maxime Geoffroy (maxime.geoffroy-@mi.mun.ca), Jonathan Fisher (Jonathan.Fisher-@mi.mun.ca), Dominique Robert (dominique_robert-@uqar.ca) | Ichthyoplankton, fish, and macrozooplankton data processed and available upon request. Acoustics, mesozooplankton, and imagery data are being processed |
| ArcticNet Seafloor Mapping Project | - | X | X | X | - | Seafloor characterization, historical sea surface condition and biological condition. Marine geohazard. Diatoms/dinoflagellate distribution. | Piston Core, Gravity Core, box cores, Seabed mapping, sub bottom profilers, phytoplankton net, ROV, MSCL, Drop Camera, Mooring | Jean-Carlos Montero-Serrano (Jeancarlos_Monteroserrano-@uqar.ca) | |
| ArcticNet-Biogeochemestry | - | X | X | X | - | Carbon Exchange Dynamics, Air-Surface Fluxes and Surface Climate : Dissolved O2/CO2, PH, Salinity, meteo-data | Underway seawater system (PCO2), Met tower, CTD-Rosette | Brent Else (belse-@ucalgary.ca) | - Being processed, but fairly far along. Could provide example figure if needed. |
| ArcticNet-Biogeochemestry | - | - | X | - | - | Underway measurements of phytoplankton productivity and trace gases: Active Chlorophyll Fluorescence, O2/N2, CH4/N2O, nutrients (NO3-) | Underway seawater system | Philippe Tortell (ptortell-@eoas.ubc.ca) | - In progress |





*Table B1 – continued from previous page*

| Programs | Leg1 | Leg2 | Leg3 | Leg4 | Leg5 | Data Collected Field of Study | Ship's Instrument | PI | Data status |
|---|---|---|---|---|---|---|---|---|---|
| Knowledge and Ecosystem Based Approach program in Baffin Bay (KEBABB) and Barrow Strait (KEBABS) (DFO) | - | - | X | - | - | – Water column biochemistry : TIC/TA, DOC/DN, Salinity, del18O, flow cytometry (FC), chlorophyll a, Phytoplankton taxonomy, Fatty acid (fA), POC/PN, FRRF  – Genomics  – Zooplankton Taxonomy and fatty Acids  – Sediment characterization  – Benthic Epifauna | – CTD-Rosette  – CTD-Rosette  – Hydrobios-TuckerNet-MonsterNet  – Box Core  – Agassiz-Beam trawl | Christine Michel (christine.michel@dfo-mpo.gc.ca) | |
| ArcticNet-Go ICE | - | - | X | - | - | Glacier Velocity and Mass Balance, Iceberg Drift Tracking | Helicopter | Luke Copland (luke.copland-@uottawa.ca), Wesley van Wychen (wesley.van.wychen-@uwaterloo.ca) | Iceberg drift data has been processed and can be included in paper; glacier velocity data won't be downloaded until summer 2022 |
| Sentinel North-Quaqtaq | - | - | X | X | - | Solenoneine analysis experiment in sediment | Box Core | Philippe Archambault (philippe.archambault-@bio.ulaval.ca) | |
| Canadian Arctic Archipelago Rivers Program (CAA-RP) and ArcticNet Contaminants and Program | - | - | X | X | - | River sampling of dissolved organic carbon (DOC), dissolved major ions (Ca, Mg, Na, K, Cl, SO4) and minor ions (Sr, Ba), stable isotopes of water (oxygen-18 and deuterium), and dissolved nutrient concentrations (Nitrate, Phosphate, Silicate) Bedload sediment and glacial till sample | Helicopter | Jean-Carlos Montero-Serrano (Jeancarlos_Monteroserrano-@uqar.ca), Kristina Brown (Kristina.Brown-@umanitoba.ca) | In progress, partial analyses completed (Nutrients, DOC, water isotopes), remaining samples archived for analyses in Spring 2023 |





*Table B1 – continued from previous page*

| Programs | Leg1 | Leg2 | Leg3 | Leg4 | Leg5 | Data Collected Field of Study | Ship's Instrument | PI | Data status |
|---|---|---|---|---|---|---|---|---|---|
| ArcticNet-NTRAIN-GEOTRACEs | - | - | X | X | - | dissolved micronutrient trace metals (Fe, Mn, Cu, Cd, Pb, Zn, Co, Ni) and Macronutrient | Go-flow (moon pool) | Jay Cullen (jcullen@uvic.ca) | In progress |
| ArcticNet-NTRAIN-Marine productivity | - | - | X | X | - | 13C/15N incub.,Nutrients, Nitrate isotopes, Ammonium, Stable isotopes, DOC/DON, POC/PN, BSi/POP, Fatty Acids, Total Lipids, Bio Markers, Taxonomy | CTD-Rosette | Jean-Éric Tremblay (jean-eric.tremblay@bio.ulaval.ca) | |
| Northern Contaminants Program (EC -Umanitoba)) | - | - | X | X | - | Microplastic and Persistent Organic pollutant (POPs) in air, water, zooplankton and sediments | CTD-Rosette, surface water (bucket), Monsternet-Tucker Net | Liisa Jantunen (liisa.jantunen@ec.gc.ca) | being pro-cessed and already pro-cessed |
| Northern Contaminants Program (EC -Umanitoba)) | - | - | - | X | - | Perfluorinated alkylated substances (PFAS), Organic contaminants (PAHs) and mercury within benthic and pelagic organisms. | Monsternet-Tucker Net - Beamtrawl | Gary Stern (gary.stern@umanitoba.ca) | Being pro-cessed |
| PECABEAU (EU-ARICE) | - | - | - | X | - | sediment and OM burial rates, concentration and composi-tion of dissolved and particu-late organic matter (DOM and POM), optical measurements (radiometry) | CTD rosette, Multicorer, Pis-ton core, Seabed mapping, MSCL, | Lisa Bröder (lisa.broeder-@erdw.ethz.ch) Michael Fritz (michael.fritz@awi.de) | In progress |





*Table B1 – continued from previous page*

| Programs | Leg1 | Leg2 | Leg3 | Leg4 | Leg5 | Data Collected Field of Study | Ship's Instrument | PI | Data status |
|---|---|---|---|---|---|---|---|---|---|
| DARKEDGE (Takuvik, Sentinel North) | - | - | - | - | X | (a) Wave, wind velocity, water temperature (b) Ice thikness (c) Near surface currents and turbulence, water temperature, relative humidity, air temperature, down-welling radiation (300-3000nm) (d) underice light properties, water column properties (e) CDOM (f) phytoplankton communities in new ice and water column (g) Condition of Calanus populations in terms of stage composition, vertical distribution, lipid content, activity patterns, and respiration rates (h) Marine productivity: Nutrients (NO3, NO2, PO4, Si), NH4, PI, Kinetic (i) Water column properties- Bioargo/float | (a) Buoys (b) Ice Canoe (c) Flux-cat (Catamaran) (d) AUV with sensors for PAR, temperature, conductivity, nitrate concentration, irradiance, chlorophyll and CDOM fluorescence, and particulate backscattering - Small ROV - C-OPS (e) CTD-Rosette (f) CTD-Rosette, phytoplankton net, microscopy, genomics, cultures (g) Hydrobios-TuckerNet-MonsterNet-IKMT-WBAT-UVP (h) CTD-Rosette (i) ARGo float equiped with CTD, Radiometer: OCR wavelengths:380, 412, 490 nm, PAR, MPE (high sensitivity) PAR, fluorescence chla, fluorescence CDOM, Backscattering, Suna (nitrates), Optode (Oxygen) -Float equiped with UVP6 | (a) Marcel Babin (marcel.babin-@takuvik.ulaval.ca) (b) Marcel Babin (marcel.babin-@takuvik.ulaval.ca) (c) Marcel Babin (marcel.babin-@takuvik.ulaval.ca) (d) Marcel Babin (marcel.babin-@takuvik.ulaval.ca) (e) Marcel Babin (marcel.babin-@takuvik.ulaval.ca) (f) Chris Bowler (cbowler-@biologie.ens.fr) (g) Malin Daase (malin.daase@uit.no), Gérald Darnis (Gerald.Darnis-@qo.ulaval.ca), Maxime Geoffroy (maxime.geoffroy-@mi.mun.ca) (h) Jean-Éric Tremblay (jean-eric.tremblay-@bio.ulaval.ca) (i) Marcel Babin (marcel.babin-@takuvik.ulaval.ca) | Data analysis in progress |
| RADCARBBS | - | - | - | X | - | Radiocarbon (Δ14C) and stable carbon (δ13C) isotopic measurements of dissolved inorganic carbon (DIC), dissolved organic carbon (DOC), and particulate organic carbon (POC) | CTD-Rosette | Brett Walker (brett.walker@uottawa.ca) | In progress, DOC/TN completed. |





*Table B1 – continued from previous page*

| Programs | Leg1 | Leg2 | Leg3 | Leg4 | Leg5 | Data Collected Field of Study | Ship's Instrument | PI | Data status |
|---|---|---|---|---|---|---|---|---|---|
| *Community of Uluhaktok (Amundsen Science)* | - | - | - | X | - | *Underwater sound ecology, wave and current regime* | *benthic tripod with current meter, CT sensor; hydrophone* | *Alexandre Forest (alexandre.forest-@as.ulaval.ca)* | |
| *ISECOLD/ISICLE* | X | - | - | - | - | *(a) Benthos characterisation: Cold water corals, sponges and invertebrates abundance, distribution, and diversity.* <br> *(b) δ13CAA and δ15N-AA signatures from sediment and zooplankton* <br> *(c) eDNA - Benthic and Pelagic Community Characterization* <br> *(d) epifaunal colonization and settlement patterns* <br> *(e) Sediment biogeochemistry and benthic pelagic nutrient coupling* <br> *(f) Mooring water and acoustic properties, organic pollutants* <br> *(g) Distribution and ecology of key pelagic species in Arctic marine food webs- Fish and zooplankton* <br> *(h) Contaminant load of fish in the Labrador Sea* | *(a) ROV video footage, gravity core, drop camera, Box Core, Baited camera* <br> *(b) boxcore, Hydrobios* <br> *(c) CTD-Rosette* <br> *(d) ROV-Box Core-array retrieval* <br> *(e) Box Core - ROV Push Core* <br> *(f) Moorings with sediment trap, hydrophone, fish tag receiver and a semi-permeable membrane device (SPMD), current meter, CT sensor, AZFP* <br> *(g) Hydrobios-IKMT-Beamtrawl- Baited remote underwater video (BRUV) camera- WBAT echosounder - EK80* <br> *(h) Hydrobios-IKMT* | *(a) David Cote (David.Cote@dfo-mpo.gc.ca), Barbara.Neves@dfo-mpo.gc.ca, Alexandre.Normandeau-@nrcan.gc.ca* <br> *(b)* <br> *(c) David Cote (David.Cote-@dfo-mpo.gc.ca)* <br> *(d) Annie Mercier (amercier@mun.ca)* <br> *(e) Chris Algar (chris.algar@dal.ca)* <br> *(f) David Cote (David.Cote-@dfo-mpo.gc.ca)* <br> *(g) Maxime Geoffroy (maxime.geoffroy-@mi.mun.ca), David.Cote@dfo-mpo.gc.ca* <br> *(h) Maxime Geoffroy (maxime.geoffroy-@mi.mun.ca), David.Cote@dfo-mpo.gc.ca* | *Data analysis in progress* |
| *CRSNG Discovery Archam-bault* | X | - | - | - | - | *Biology of shrimp in the Arctic (Shrimp's diet and eggs)* | *IKMT* | *Philippe Archambault(philippe.archambault @bio.ulaval.ca) Maxime Geoffroy (maxime.geoffroy-@mi.mun.ca), Guillaume Blais (guillaume.blais.8-@ulaval.ca)* | *Data analysis in progress* |





## Appendix C: Additional figures

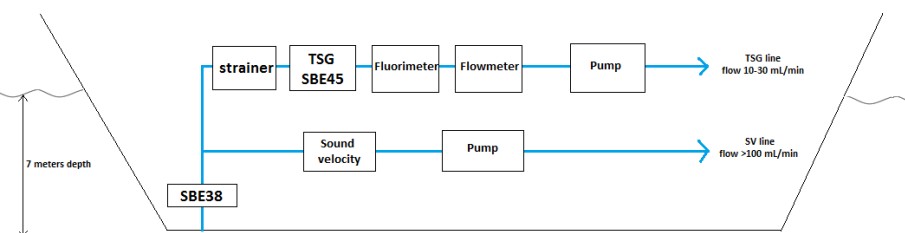

**Figure C1.** Thermosalinograph Schematic

Schematic of the TSG

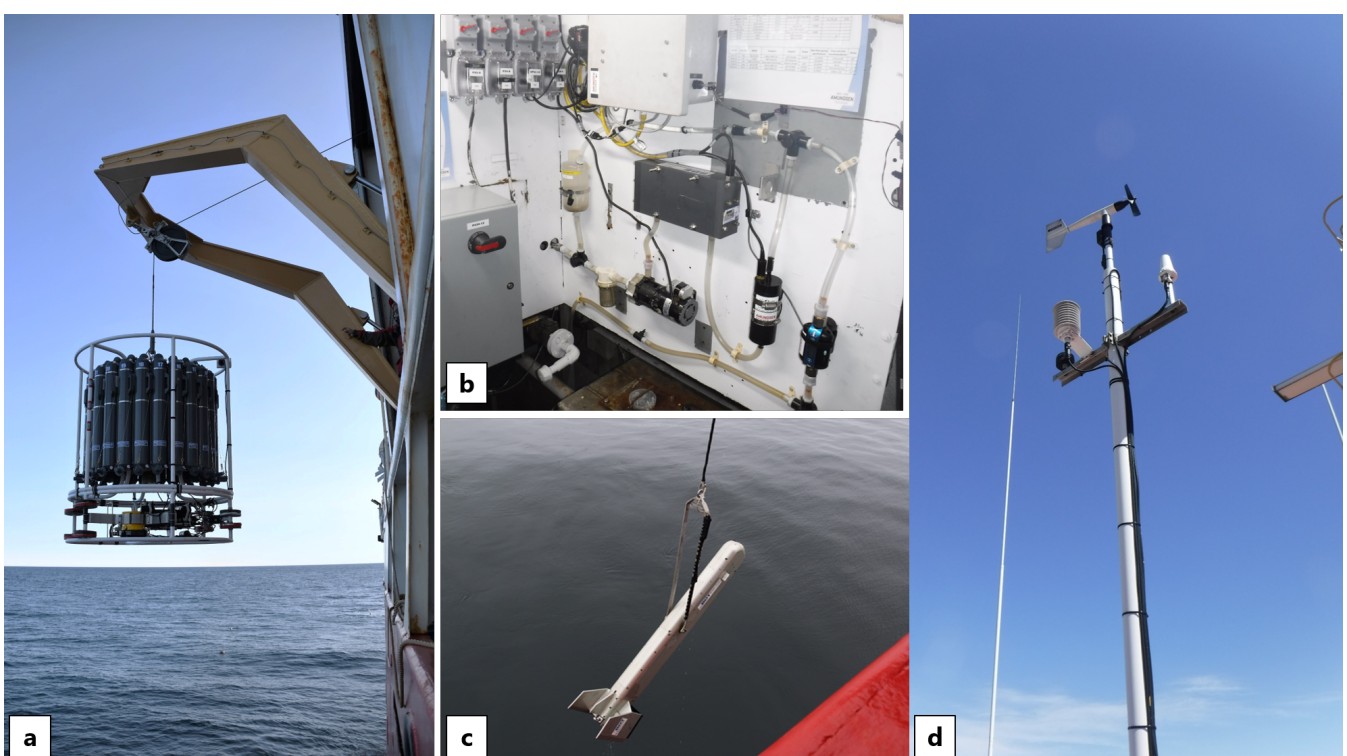

**Figure C2.** Photo of (a) the CTD Rosette being deployed with the A-Frame, (b) the TSG in the enfgine room, (c) the MVP fish being deployed and (d) the AVOS system.

*Author contributions.* These authors contributed equally to this work.



*Acknowledgements.* We thank the commanding officers and crews of the CCGS *Amundsen* for providing support during the entire 2021 ex-
385   pedition. This sampling could not have been undertaken without the interest and contribution of the scientific community and user programs.
We thank Philippe Massicotte (Université Laval) for providing suggestions. This work was supported by the Major Scientific Infrastructure
fund of the Canada Foundation for Innovation.





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
