# Peer review of "Oceanographic dataset collected during the 2021 scientific expedition of the Canadian Coast Guard Ship *Amundsen"

_Earth System Science Data, 2023_

## Referee Comment (RC1)

**Comment on essd-2023-204 by Anonymous Referee**

**General Comments**

The authors describe an extensive dataset of sea surface, atmospheric and seabed underway measurements collected in the Canadian Arctic waters during the 2021 scientific expedition performed by the research icebreaker Amundsen.
The processing methods and an overview of the data collected during that expedition are presented as well.

These data are particularly precious and relevant as they are part of a major long-term monitoring program of the area, started in 2003, to improve the understanding and the study of the unique processes taking place in the Canadian and Greenlandic Arctic.
Moreover, these observations can be useful to guide local decision makers to monitor risks associated to different activities (e.g., exploration and exploitation of oil and gas) and potentially beneficial for other ecological, geomorphological, sedimentological and management applications. Hence, this study certainly meets the ESSD criteria for data availability.

The presentation of the study sites and instruments is detailed. Figure 1 could be improved by the addition of general hydrodynamic features of the study area, based on available references.

Reference is made to hydrodynamic sensors and data, but no data is introduced and discussed.

In general, it may be useful to provide references on where the data and results of previous expeditions are available. The present manuscript is only focused on the 2021 expedition, in line with ESSD journal, but some long-term comments and/or comparison/analysis (e.g., studies of regional trends and variability) with respect to previous expedition could be shown here or in a future dedicated work.

**Specific Scientific Comments**

Lines 26-28: Please add some details and reference about the referred unique ecosystems of the Arctic Ocean.

Lines 32-35: Reference is made to more than 1400 refereed scientific publications and 400 datasets resulting from CCGS Amundsen expeditions since 2003. Are these also publicly available on an annual basis? If yes, please add at least the most recent ones to the reference list or add a reference to where this information can be found.

Lines 41-42: Please add labels for Labrador Sea, the Baffin Bay, the Canadian Arctic Archipelago, and the Beaufort Sea in Figure 1. Also consider adding the main circulation features as detailed in Section 1.1, distinguishing between cold and warm waters.

Line 144: Please add reference to the Winkler method.

Line 151: Please add reference to the Seabird website or to the Seabird data processing software.

Section 2.2: Were collocated data from the CTD-Rosette and in-situ samples used for inter-comparison and correction of the Moving Vessel Profiler data, where available?

Line 254: If TS Diagram represented in Figure 5 are derived by transect profiles during leg 3 of 2021 expedition, it is not clear what is derived from Curry et al. (2011); Tang et al. (2004) in Fig. 5.

Figure 4, 5, 9: It is not clear what isolines represent as regards panel b.

Line 290: When you report: "Such dataset can be compare to the ones collected by the Amundsen during the 2021 expedition", Do you mean that the 2 datasets are comparable or that you are going to compare them in a future works? If you intended the former, please rephrase and show results or comments on this.

Line 295: "Only the first 148 casts are presented in this paper". Please explain.

**Technical Comments**

In general, along the entire manuscript, please be consistent and consider that the symbol for the unit shall be placed after the numerical value in the expression for a quantity, with a thin space between them, e.g., 12 m, 10 °C.

Line 25: "1.5-3.7 °C" instead of the italics "1.5-3.7 °C".

Line 48: "contacting the respective principal investigators (PI, as detailed in Table B1)", instead of "contacting the respective principal investigators (as detailed in Table B1)".

Page 35 - Table B1: Adjust the length of the last 2 columns to correctly display the terms in the last row.

Line 112: Add the full stop at the end of the sentence.

Line 128: Please modify: "are publicly available, after registration, at Guillot et al. (2022)" instead of "are publicly available at Guillot et al. (2022)".

Line 149: "faulty sensor" instead of "faulty senor".

Line 192 - Figure 3: Please use lowercase letters (a, b,...) to label parts of the figure (both in the image, in its caption and in the main text).

Line 236: Spatially/Vertically interpolated?

Lines 242-244 "The Davis Strait water masses have been extensively described in the past (Tang et al., 2004; Curry et al., 2011; Lehmann et al., 2019; Punshon et al., 2014) and the four main water masses of the region were observed (Fig. 5) during the sampling from the CCGS Amundsen" instead of "The Davis Strait water masses have been extensively described in the past and the four main water masses of the region were observed (Fig. 5) during the sampling from the CCGS Amundsen (Tang et al., 2004; Curry et al., 2011; Lehmann et al., 2019; Punshon et al., 2014)".

Lines 246, 248, 252, 262, 265, 283, 284: "µM" instead of the italics "*µM*".

Line 253: "Fig. 4b" instead of "Fig. 9b".

Line 258: "at each stations along the NOW transect." instead of "at each stations along the NOW transect, ."

Figure 4, 5, 9: "Isolines represent" instead of "Isolines represents".

Line 265: "DO, ranging from 295-340 µM" instead of "DO,ranging from 295-340*µM*"

Line 270: "TS-diagram (Figure 7). Different" instead of "TS-diagram (Figure 7).Different"

Lines 271-272: Please check this sentence: "Along the transect, two water masses defined by two sets of conservative temperature, on the left side of the figure".

Figure 6 (d) and 7: Check the labelling of the extremes of the colour scale.

Line 289: "in order to facilitate their reuses" instead of "in order facilitate their reusise".

Figure 10: Please add to the caption: "Isolines represent conservative temperature (isotherms, in °C)". Please add a panel about conservative temperature if possible.

Line 311: AS acronym for absolute salinity already defined.

Figure 12: Please add unit in the legend.

Line 366: Please correct the italics.

Line 376: Link is missing.

Figure C2: Consider moving this Figure to the main text.

---

## Author Response (AR1)

**Answers to reviewer #1**

- ✓ **Lines 26-28: Please add some details and reference about the referred unique ecosystems of the Arctic Ocean.**

We did not add details, the main point are already in the sentence. Adding information would be outside of the scope of the paper.

- ✓ **Lines 32-35: Reference is made to more than 1400 refereed scientific publications and 400 datasets resulting from CCGS Amundsen expeditions since 2003. Are these also publicly available on an annual basis? If yes, please add at least the most recent ones to the reference list or add a reference to where this information can be found.**

Datasets resulting from the central pool of scientific instruments onboard the Canadian Coast Guard Ship Amundsen since 2003 are available for public access at the Polar Data Catalogue (https://www.polardata.ca/ ) and other websites; for instance the bathymetry data are accessible at http://www.omg.unb.ca/arctic-mapping/ ; https://www.ncei.noaa.gov/maps/bathymetry/ ;

The Amundsen Science website includes an overview of all the available data at the Polar Data Catalogue (https://amundsenscience.com/data/data-access/).

We keep track of the scientific publications resulting from the CCGS Amundsen expeditions for reporting and internal purposes only. While we do not have a list of publications publicly available, we referred a few recent publications at line 32-35. The publications range from physical oceanography to geology, biogeochemistry, ecology and safety hazard assessments.

- ✓ **Lines 41-42: Please add labels for Labrador Sea, the Baffin Bay, the Canadian Arctic Archipelago, and the Beaufort Sea in Figure 1. Also consider adding the main circulation features as detailed in Section 1.1, distinguishing between cold and warm waters.**

Main circulations and labels were added in Figure 1 as suggested.

- ✓ **Line 144: Please add reference to the Winkler method.**

The following reference will be added.

Aminot, A., & Chaussepied, M. (1983). Manuel des analyses chimiques en milieu marin. Editions Jouve, CNEXO, Paris, 395 p.

✓ **Line 151: Please add reference to the Seabird website or to the Seabird data processing software.**

The following reference will be added.

Sea-Bird Electronics, Inc. (n.d): http://seabird.com/software/sbe-data-processing

✓ **Section 2.2: Were collocated data from the CTD-Rosette and in-situ samples used for inter-comparison and correction of the Moving Vessel Profiler data, where available?**

Collocated data from the CTD Rosette were available and used for inter-comparison and correction of the MVP Profiler data. In-situ samples were used to validate data from the CTD-Rosette. More details of the processings are available in the report processing (available with the data).

✓ **Line 254: If TS Diagram represented in Figure 5 are derived by transect profiles during leg 3 of 2021 expedition, it is not clear what is derived from Curry et al. (2011); Tang et al. (2004) in Fig. 5.**

The sentence in Line 254 was reformulated as follow:

The distinction between the different water masses observed in Fig. 5 has been made according to Curry et al. (2011); Tang et al. (2004).

✓ **Line 290: When you report: "Such dataset can be compare to the ones collected by the Amundsen during the 2021 expedition", Do you mean that the 2 datasets are comparable or that you are going to compare them in a future works? If you intended the former, please rephrase and show results or comments on this.**

Line 290 now reads:

The Amundsen has been sampling in the Cape Bathurst area 12 times since 2003, which produces an extensive dataset that could be used by readers to assess interannual variability and trend in the region.

Recently, Massicotte et al. (2021) have compiled and standardized the collected datasets from the 2009 MALINA expedition in the Beaufort Sea in order facilitate their reuses.

✓ **Line 295: "Only the first 148 casts are presented in this paper". Please explain.**

After the 148th dive of the MVP, the ship has changed its heading to enter Diana Bay towards Quaqtaq. In order to make sense of the transect, only profiles acquired in a straight line (while crossing Hudson Strait) are presented in the paper and in figure.

**Answers to reviewer #2**

- ✓ **Fig.4 6 9 10 I doubt that temperature isolines reported in salinity maps and salinity isolines in temperature maps can help for a better data interpretation. I would suggest to use the same isolines of the displayed parameters or to better explain this in the figure captions.**

While we were hoping to maximize the information presented on each figure by adding isolines from different variables other than the ones displayed in color, we realize that it complicates the figure. and on the figures to maximize the information presented for better interpretation. Actually it does not, so we decided to follow your recommendation on using isolines from the displayed variable. It appears to help reading the figure.

- ✓ **2.1 What is the depth of TSG intake?**

The depth of TSG intake is 7 m.

- ✓ **2.2 Which was the maximum vessel speed to operate the MVP ?**

The maximum vessel speed is 8 knots

- ✓ **In Fig 10 Use the same units of the others figures for DO. Also report the temperature map. Are space and time scale consistent with the presence of internal waves?**

The unit of the DO on the Fig10 is now converted into microMolar.

We believe they are not related to the presence of internal wave. They seem to be generated by the repetitive free fall motion of the MVP fish along the transect. However, it is still premature to draw conclusions on the nature of the real phenomenon behind such structures. We did not comment on them to avoid interpretations of the data in order to meet a Data Paper content expectations. In this transect, the distances between each MVP free fall is generally about 500m, sometimes reaching 1km depending on depth of the water along the transect.

- ✓ **LADCP is mentioned (line 137) but nothing is shown or commented. Why ?**

LADCP is only mentioned as an example of what kind of instruments are installed on the Rosette, the same way other instruments such as the Deep Suna (Nitrate sensor) are presented in Table 1. The paper focuses mainly on the 2021 data expedition. However, LADCP data since 2010 are available freely for public access at Polar Data Catalogue under CCIN 12714 or with this link https://www.polardata.ca/pdcsearch/PDCSearchDOI.jsp?doi_id=12714

---

## Author Response (AR2)

**Author's Responses**

According to the topic editor, some corrections are necessary before the manuscript can be published. The following adjustments were applied to the manuscript to produce the final version submitted.

**lines 36-37: please list in alphabetical order as all references are from 2023**

The indicated references in Lines 36-37 are listed in alphabetical order.

**lines 254-255: please invert references starting from the older**

The references in Lines 254-255 are inverted starting the older.